# Offline Model-based Adaptable Policy Learning

**Xiong-Hui Chen**[1], **Yang Yu**[1,3,*], **Qingyang Li**[2], **Fan-Ming Luo**[1], **Zhiwei Qin**[2],
**Wenjie Shang**[2], **Jieping Ye**[2]

[1] National Key Laboratory of Novel Software Technology, Nanjing University, Nanjing, China
[2] AI Labs, Didi Chuxing
[3] Polixir.ai
chenxh@lamda.nju.edu.cn, yuy@nju.edu.cn, qingyangli@didiglobal.com
luofm@lamda.nju.edu.cn
{qinzhiwei,shangwenjie,yejieping}@didiglobal.com

## Abstract

In reinforcement learning, a promising direction to avoid online trial-and-error costs is learning from an offline dataset. Current offline reinforcement learning methods commonly learn in the policy space constrained to in-support regions by the offline dataset, in order to ensure the robustness of the outcome policies. Such constraints, however, also limit the potential of the outcome policies. In this paper, to release the potential of offline policy learning, we investigate the decision-making problems in out-of-support regions directly and propose offline Model-based Adaptable Policy LEarning (MAPLE). By this approach, instead of learning in in-support regions, we learn an adaptable policy that can adapt its behavior in out-of-support regions when deployed. We conduct experiments on MuJoCo controlling tasks with offline datasets. The results show that the proposed method can make robust decisions in out-of-support regions and achieve better performance than SOTA algorithms.

## 1 Introduction

Recent studies have shown that reinforcement learning (RL) is a promising approach for real-world applications, e.g., sequential recommendation systems [1, 2, 3, 4] and robotic locomotion skill learning [5, 6]. However, the trial-and-error of RL in the real world [7] obstructs further applications in cost-sensitive scenarios [8].

Offline (batch) RL learns a policy within a static dataset collected by a behavior policy without additional interactions with the environment [8, 9, 10, 11]. Since it avoids costly trial-and-error in real-world environments, offline RL is a promising way to handle the challenge in cost-sensitive applications. A significant challenge of offline RL is in answering counterfactual queries, which asks about how the performance (e.g., Q value) would have been if the agent were to execute an unseen action sequence, then learning to make optimal decisions based on the performance [8]. Fujimoto et al. [10] have shown that the distributional shift of states and actions, which comes from the discrepancy between evaluated policies and behavior policies, often leads to large extrapolation error in value function estimation. In traditional model-free algorithms, the extrapolation error in value function estimation hurts the generalization performance of the learned policies. Since the additional samples, which can correct value estimation errors, are unavailable in the offline setting, the performance of learned policies based on value function is unstable [10].

On the other hand, model-based RL techniques, which learn dynamics models from collected datasets and learn the value function and policies based on the dynamics models, do not need to estimate

---

*Corresponding author

35th Conference on Neural Information Processing Systems (NeurIPS 2021).

the value functions rely on the collected datasets. However, similar challenges occur in dynamics model approximation. The dynamics model might overfit the limited dataset and suffer extrapolation errors in regions that behavior policies have not visited, which causes instability of the learned policy when deployment [12]. Here we call it out-of-support regions. Moreover, in model inference, the compounding error, that is, the accumulated prediction errors between simulation trajectories and reality, would be large even if the one-step prediction error is small [13, 14]. Recent studies in offline model-based RL [12, 15] have made significant progress in MuJoCo tasks [16]. These methods constrain policy sampling in dynamics models for robust policy learning. By using large penalty [12] or trajectory truncation [15, 12] in the regions with large prediction uncertainty (uncertainty is a designed metric to evaluate the confidence of prediction correctness) or compounding error, policy exploration is constrained in the regions of dynamics models where the predictions are corrected with high confidence, so as to avoid exploiting regions with risks of large extrapolation error. However, the constraints on dynamics models lead to a conservative policy learning process, which limits the potential of leveraging dynamics models: The visits on states and actions in out-of-support regions are more likely to be inhibited by the constraints, making the learned policy restrict the agent to be in similar regions as the behavior policy.

From the perspectives of counterfactual queries, we consider that model-based RL is promising to handle offline RL — ideal reconstructed dynamics models can simulate the transition dataset without the distributional-shift problem given any policy, and the value function can be estimated via the "simulated" transition dataset directly. The bottleneck of offline model-based RL comes from the policy learning in the approximated dynamics model with extrapolation error. In this paper, instead of learning by tightly constraining policy exploration in in-support regions, we investigate decision-making in out-of-support regions directly. Finally, we propose a new offline policy learning framework, offline Model-based Adaptable Policy LEarning (MAPLE), to address the aforementioned issues. Ideally, MAPLE tries to model all possible transition dynamics in the out-of-support regions. Then an Adaptable policy is learned to be aware of each case to adapt its behavior to reach optimal performance. In the practical version of MAPLE, we use an ensemble technique to construct ensemble dynamics models. To be aware of each case of the transition dynamics and learn an adaptable policy, we use a meta-learning technique that introduces an extra environment-context extractor structure to represent dynamics patterns, and the policy adjusts itself according to the environment contexts.

We conduct experiments on the MuJoCo tasks. The results show that the sampling regions for robust offline policy learning can be extended via constructing transition patterns in out-of-support regions to cover the real case. The output adaptable policy yields better performance than SOTA algorithms when deployed. MAPLE gives a new direction to handle the offline policy learning problem in the dynamics models: Besides constraining on sampling and training dynamics models with better generalization, we can also model out-of-distribution regions by constructing all possible transition patterns.

## 2 Related Work

Reinforcement learning (RL) has shown to be a promising approach to complex real-world decision-making problems [1, 2, 3, 4]. However, unconstrained online trial-and-error in the training of RL agents prevents further applications of RL in safety-critical scenarios since it might result in large economic losses [8, 17, 18, 19]. Many studies propose to overcome the problem by offline (batch) RL algorithms [20]. Prior works on offline RL are based on model-free algorithms. To overcome the extrapolation error, which is introduced by the discrepancy between the offline dataset and true state-action distribution of learned target policies [10], these methods are designed to constrain target policies to be close to the behavior policies [10, 11, 21], to apply ensemble methods for robust value function estimation [22], or to re-weight samples in datasets with importance sampling [23]. Most recent studies have shown that policy learning with an approximated dynamics model has good potential to take robust actions outside the action distribution of behavior policies [15, 12]. The challenge comes from the extrapolation error of the dynamics models in out-of-support regions. To address the issues, these methods learn policies from dynamics models with uncertainty constraints. Uncertainty is a measure of prediction confidence on next states. The uncertainty is often computed by the inconsistency in the ensemble dynamics model predictions for each state-action pair. Kidambi et al. [15] construct terminating states based on a hard threshold on uncertainty, while Yu et al. [12] use a soft reward penalty to incorporate uncertainty. The penalty constrains policy exploration and

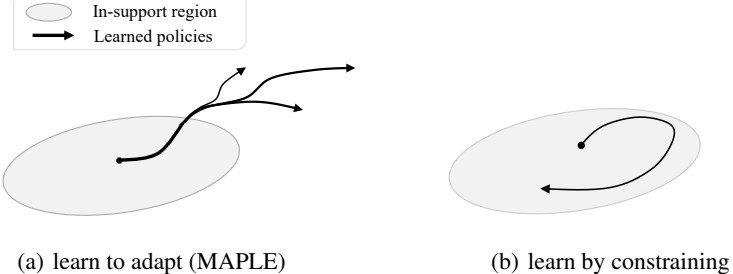

(a) learn to adapt (MAPLE)      (b) learn by constraining

Figure 1: Illustration of MAPLE compared with learning by constraining. The pointed lines represent the optimal trajectories of the learned policies. There are several policies in MAPLE since the method learns to adapt to multiple dynamics models. The gray oval represents the in-support region.

optimization to the regions with high consistency for better worst-case performance in the deployment environment [15, 12].

The difference between the aforementioned model-based methods and MAPLE is shown in Figure 1. Compared with previous methods [15, 12] learned by constraining (in Figure 1(b)), we learn to adapt to all possible dynamics transitions in the states in out-of-support regions (in Figure 1(a)).

## 3 Background and Notation

In the standard RL framework, an agent interacts with an environment governed by a Markov Decision Process (MDP) [24]. The agent learns a policy $\pi(a_t|s_t)$, which chooses an action $a_t \in \mathcal{A}$ given a particular state $s_t \in \mathcal{S}$, at each time-step $t \in \{0, 1, ..., T\}$, where $T$ is the trajectory length. $\mathcal{S}$ and $\mathcal{A}$ denote the state and action spaces, respectively. The reward function $r_t = r(s_t, a_t) \in \mathbb{R}$ evaluates the immediate performance of the action $a_t$ given the state $s_t$. The goal of RL is to find an optimal policy $\pi^*$ which maximizes the multi-step cumulative discounted reward (i.e., long-term performance). The objective of RL is $\max_\pi J_\rho(\pi) := \mathbb{E}_{\tau \sim p(\tau|\pi,\rho)}\left[\sum_{k=0}^{T} \gamma^k r_k\right]$, where $\gamma$ is the discount factor, and $p(\tau|\pi, \rho)$ is the probability of generating a trajectory $\tau := [s_0, a_0, ..., a_{T-1}, s_T]$ under the policy $\pi$ and a dynamics model $\rho(s_{t+1}|s_t, a_t)$. In particular, $p(\tau \mid \pi) := d_0(s_0) \prod_{t=0}^{T-1} \rho(s_{t+1} \mid s_t, a_t)\pi(a_t|s_t)$, where $d_0(s_0)$ is the initial state distribution. A common way to find an optimal policy $\pi^*$ is to optimize the policy with gradient ascent along $\nabla J_\rho(\pi)$ [24, 25].

In the offline RL setting, we are given only a static dataset $\mathcal{D} = \{(s_i, a_i, r_i, s_{i+1})\}$ collected by some unknown policy. The goal is to obtain a policy that maximizes $J_\rho$ by only using the static dataset.

## 4 Method

We argue that in current offline model-based methods, the constraints of sampling to in-support regions of the dynamics model lead to a conservative policy learning process, limiting the potential of leveraging dynamics models. In this paper, to relax the constraints on the dynamics model, we investigate decision-making in out-of-support regions directly.

In this section, we first give a motivating example to show our ideal solution to out-of-support region decision-making (in Section 4.1). Then, we introduce a practical algorithm of the proposed solution for complex tasks, based on meta-learning techniques (in Section 4.2 and Section 4.3).

### 4.1 Decision-Making in Out-of-Support Regions

By rethinking the scheme of offline model-based RL, without loss of generality, we first formulate the problem as decision-making with a partially known dynamics model (Pak-DM) in a surrogate objective. In this problem, we have two dynamics models: a target dynamics model $\rho$ and a partially known dynamics model $\rho'$, where $\rho$ is the deployment environment in the offline RL setting, and $\rho'$ is used to approximate $\rho$. Due to the bias of data sampling in the offline setting and the

limitation on the capacity of the function approximator, only in part of state-action space, we have $\rho'(s'|s,a) = \rho(s'|s,a)$, while the transitions in other parts of space are uncertain. We call the satisfied space "accessible space" (a.k.a., in-support regions) and its complement "inaccessible space" (a.k.a., out-of-support regions). In this problem, we assume the two subspaces have been predefined in some ways as in the offline setting (e.g., we can define the space in which an uncertainty quantification is larger than a threshold as the inaccessible space) and then discuss the decision-making problem in this setting. Formally, given an accessible space $\mathcal{X}_a$ and an inaccessible space $\mathcal{X}_i$, the partially known dynamics model is defined as:

$$\rho'(s'|s,a) = \begin{cases} \rho(s'|s,a) & [s,a] \in \mathcal{X}_a \\ \text{Unknown} & [s,a] \in \mathcal{X}_i \end{cases},$$

where $\mathcal{X}$ denotes the state-action concatenated space for brevity and $[s,a]$ denote a vector concatenating $s$ and $a$. Our objective is to find a policy $\pi^*$ to maximize $J_\rho$ by only querying the partially known dynamics model $\rho'$. If $\mathcal{X}_i = \emptyset$, that is $\rho'(s'|s,a) = \rho(s'|s,a), \forall s \in \mathcal{S}, a \in \mathcal{A}$, the problem is reduced to a vanilla model-based policy learning problem with an oracle dynamics model. For simplification, we assume the oracle reward function $r$ is given, but it can also be formulated as a partially known reward function in a similar way. Now we give an example of a Pak-DM in Figure 2.

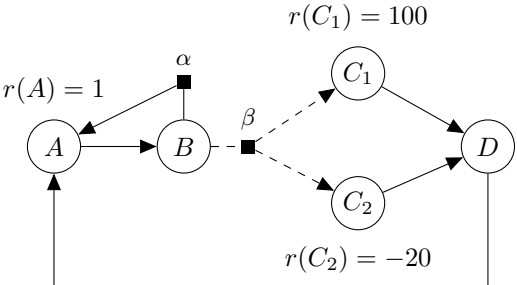

Figure 2: An example of a Pak-DM. Each node denotes a state. Here we consider an MDP with finite state space, including $A$, $B$, $C_1$, $C_2$ and $D$. The directed edges denote the transition process. Here we consider a one-action transition in all states except for state $B$. On state $B$, we have action $\alpha$ and $\beta$, which are denoted as square nodes. By taking $\alpha$ on state $B$, the state will change to $A$. However, the transition after taking $\beta$ in $B$ is unknown. We use dashed directed edges to denote the possible transitions. In our formulation, edges to any node are valid. But we just consider the edges from $B$ to $C_1$ and $C_2$ and omit the edges to $A$, $B$, and $D$ for better readability. The reward function $r(s)$ will give a reward when the agent reaches $A$, $C_1$ or $C_2$.

In this setting, the model-based policy learning algorithms with constraints ([12, 15]) can be summarized as: finding the optimal policy without reaching inaccessible space. It might output a conservative policy since it avoids making decisions that might lead agents to out-of-support regions. Taking Figure 2 as an example, the output policy would be run in the loop of $A \to B \to \alpha \to A$. It will avoid taking $\beta$ on state $B$. If the real transition is $\rho(B, \beta) = C_2$, the policy can avoid the penalty $-20$ since $C_2$ will not be reached. On the other hand, while $\rho(B, \beta) = C_1$, the policy would miss the large bonus 100 in $C_1$.

Our question is, how should we make good decisions in out-of-support regions directly so that we can make better use of the approximated $\rho'$ for better performance? We give a new paradigm to solve the problem which is the ideal implementation version of MAPLE:

1. (Training) Construct a dynamics model set $\{\hat{\rho}_i\}$ via modeling all possible transitions in $\mathcal{X}_i$. (It is impractical to do that with infinite state space. We will give a practical solution in Section 4.3);

2. (Training) Learn the optimal policy from each model $\hat{\rho}_i$ to form the optimal policy set $\{\pi^*_{\hat{\rho}_i}\}$;

3. (Deployment) Initialize a state $s_0$ from the deployment environment $\rho$;

4. (Deployment) *Probe* the environment $\rho$ by selecting an action $a$ such that $[s,a] \in \mathcal{X}_i$. After getting the next state $s' = \rho(s'|s,a)$, store the tuple $(s, a, s')$ in a memory (e.g., a replay

buffer) $\mathcal{D}$. If there is no action $a$ that allows $[s, a] \in \mathcal{X}_i$, randomly select a policy from the policy set to take an action.

5. (Deployment) *Reduce* the policy set by only keeping the policies whose corresponding transition model $\hat{\rho}_i$ can explain the experiences in the memory: $\{\hat{\rho}_i\} \leftarrow \{\rho \mid \rho(s'|s, a) = s', \forall(s, a, s') \in \mathcal{D}, \forall \rho \in \{\hat{\rho}_i\}\}$ and $\{\pi^*_{\hat{\rho}_i}\} \leftarrow \{\pi^*_\rho | \rho \in \{\hat{\rho}_i\}\}$;

6. (Deployment) Repeat Step 4 and 5 until the policy set is reduced to a single policy.

In this paradigm, we solve the decision-making problem in out-of-support regions by probing the uncertainty part of the deployment environment and adapting the policy for the environment. In Figure 2, the ideal MAPLE solution would construct two dynamics models: $\hat{\rho}_1$ where $\hat{\rho}_1(B, \beta) = C_1$, and $\hat{\rho}_2$ where $\hat{\rho}_2(B, \beta) = C_2$. Then we learn two optimal policies $\{\pi^*_{\rho_1}, \pi^*_{\rho_2}\}$ for each dynamics model. At deployment, we first randomly select a policy from the policy set to make decisions. Upon reaching $B$ for the first time, where the transition on action $\beta$ is uncertain, we take action $\beta$ and get the next state. If the next state is $C_1$, the policy will reduce to $\pi^*_{\hat{\rho}_1}$, otherwise to $\pi^*_{\hat{\rho}_2}$. Therefore, if $\rho(B, \beta) = C_2$, the policy would initially run $A \rightarrow B \rightarrow \beta \rightarrow C_2 \rightarrow D \rightarrow A$ and then run in the loop of $A \rightarrow B \rightarrow \alpha \rightarrow A$ because the latter yields higher rewards. If $\rho(B, \beta) = C_1$, the policy would always run in the loop of $A \rightarrow B \rightarrow \beta \rightarrow C_1 \rightarrow D \rightarrow A$.

We then dive into the performance difference of the two paradigms for decision-making. Formally, we give Theorem 1 to describe it. The full proof can be found in Appendix A.

**Theorem 1** *Given a target dynamics model $\rho$, a policy $\pi_a$ learned by adapting, a policy $\pi_c$ learned by constraints, and the maximum step $N_m$ taken by $\pi_a$ to probe and reduce the policy set to a single policy, the performance gap between $\pi_a$ and $\pi_c$ satisfies:*

$$J_\rho(\pi_a) - J_\rho(\pi_c) \geq \Delta_c - \Delta_p - \gamma^{N_m+1} J_{\rho_\Delta}(\pi^*),$$

*where $\Delta_c$ denotes the performance gap of the optimal policy $\pi^*$ and $\pi_c$, while $\Delta_p$ denotes the performance degradation of MAPLE compared with $\pi^*$ because of the phase of probing. $J_{\rho_\Delta}(\pi^*)$ denotes the performance degradation of $\pi^*$ on the dynamics model $\rho$ caused by different initial state distribution: $J_{\rho_\Delta}(\pi^*) = \mathbb{E}_{d^{\pi^*}_{N_m+1}(s)}[V^\star(s)] - \mathbb{E}_{d^{\pi_a}_{N_m+1}(s)}[V^\star(s)]$, where $d^{\pi^*}_{N_m+1}(s)$ and $d^{\pi_a}_{N_m+1}(s)$ denote the state distribution induced by $\pi^\star$ and $\pi_a$ at the $N_m+1$ step and $V^*(s)$ denotes the expected long-term rewards of $\pi^\star$ at state $s$.*

We can see that the performance gain of $\pi_a$ is that it can automatically converge to the optimal policy after the loop of probing and reducing (i.e., $\Delta_c$), while the cost of $\pi_a$ comes from additional probing on inaccessible space, including less reward getting when probing (i.e., $\Delta_p$) and a worse initial state distribution after probing (i.e., $J_{\rho_\Delta}(\pi^*)$).

Based on Theorem 1, we give the principles for choosing between the paradigms: Firstly, with a larger performance gap of $\Delta_c$, $\pi_a$ can reach a better performance than $\pi_c$. On the other hand, the tasks with large penalties on undesired behavior might make $\Delta_p$ larger, which reduces the overall performance of $\pi_a$. Besides, the tasks where sub-optimal behaviors easily lead agents to states with low value, e.g., unsafe states which are prone to terminate the trajectory, might make $J_{\rho_\Delta}(\pi^*)$ large, which also reduces the overall performance of $\pi_a$.

### 4.2 Efficient Decision-Making in Out-of-Support Regions with Meta-learning Techniques

It is computationally inefficient to learn optimal policies independently for each dynamics model since the policies' behaviors would be similar in in-support regions. For better efficiency, we introduce a context-aware adaptable policy, inspired by meta-learning techniques, to represent the set of learned policies. Here, we first introduce a new notation: the environment-context vector $z \in \mathcal{Z}$, where $\mathcal{Z}$ denotes the space of the context vectors. Given a set of dynamics models $\mathcal{T} := \{\hat{\rho}_i\}$, each $\hat{\rho}_i$ can be represented by a vector $z$. Formally, there is a mapping $\phi : \mathcal{T} \rightarrow \mathcal{Z}$. We call $\phi$ an environment-context extractor. The context-aware policy $\pi(a|s, z)$ takes actions based on the current state $s$ and the vector of environment-context $z$ of a given environment $z = \phi(\hat{\rho})$, where $\hat{\rho} \in \mathcal{T}$. We define the optimal environment-context extractor $\phi^*$ to satisfy: $\exists \pi_{\phi^*} \in \Pi, \forall \hat{\rho} \in \mathcal{T}, J_{\hat{\rho}}(\pi_{\phi^*}) = \max_\pi J_{\hat{\rho}}(\pi)$, where $\pi_\phi := \pi(a_t|\phi(z_t|\rho_t), s_t)$ is an adaptable policy and $\Pi$ denotes the policy class. We discuss the input of $\phi$ later. In addition, we define the optimal adaptable policy $\pi^*_{\phi^*}$ to be one that satisfies $\forall \hat{\rho} \in \mathcal{T}, J_{\hat{\rho}}(\pi^*_{\phi^*}) = \max_\pi J_{\hat{\rho}}(\pi)$. With the optimal $\phi^*$ and $\pi^*_{\phi^*}$, and given any $\hat{\rho}$ in $\mathcal{T}$, the adaptable

policy can make the best decisions with the output of environment-context $z$. To achieve this, given a dynamics model set $\mathcal{T}$, we optimize $\phi$ and $\pi_\phi$ by the following objective function:

$$\phi^*, \pi_{\phi^*}^* = \arg\max_{\phi, \pi_\phi} \mathbb{E}_{\rho \sim \mathcal{T}} \left[ J_\rho(\pi_\phi) \right], \tag{1}$$

where $\sim$ denotes a sample strategy to draw dynamics models $\hat{\rho}$ from the dynamics model set $\mathcal{T}$ s.t. $P[\hat{\rho}] > 0, \forall \hat{\rho} \in \mathcal{T}$. We take a uniform sampling strategy in the following analysis.

To learn the context $z$ by $\phi(z|\hat{\rho})$, the main question is: What are suitable inputs to $\phi$ for context learning? In the robotics domain, similar environment contexts have been proposed recently [26, 27, 28]. The policy incorporates an online system identification module $\phi(z_t|s_t, \tau_{0:t})$, which utilizes the history of past states and actions $\tau_{0:t} = [s_0, a_0, ..., s_{t-1}, a_{t-1}, s_t]$ to predict the parameters of the dynamics in simulators. For example, $\tau$ could be a trajectory of robot interactions with varying friction coefficients, and $z$ is the value of the coefficient. In practice, a recurrent neural network (RNN) is used to embed the sequential information into environment-context vectors $z_t = \phi(s_t, a_{t-1}, z_{t-1})$. In MAPLE, we follow the same structure to model the extractor but the trajectories are rollout in the constructed dynamics models. If the reward function is also partially known, $r_{t-1}$ should be considered, that is $z_t = \phi(s_t, a_{t-1}, r_{t-1}, z_{t-1})$.

With an RNN-based environment-context extractor $\phi$ optimized with Equation (1), the context-aware policy $\pi$ can automatically probe environments and reduce the policy set. Considering a given $i$-step partial trajectory $\tau_{0:i}$ and a subset of deterministic dynamics models $\mathcal{T}' \subset \mathcal{T}$ where the dynamics model $\hat{\rho} \in \mathcal{T}'$ is consistent with $\tau_{0:i}$, the objective from $i+1$ to $T$ on given $\tau_{0:i}$ can be rewritten as $\mathbb{E}_{\hat{\rho} \sim \mathcal{T}'}[\mathbb{E}_{\tau \sim p(\pi, \hat{\rho})}[\sum_{k=i+1}^{T} \gamma^k r_k]]$. Since the dynamics models in $\mathcal{T}'$ are indistinguishable at step $i+1$, the optimal policy at this step would converge to a stochastic policy if the optimal actions are different among the dynamics models. If $\hat{\rho}$ is sampled uniformly from $\mathcal{T}'$ and the optimal cumulative rewards $\sum_{k=i+1}^{T} \gamma^k r_k^*$ are the same for each dynamics model, the optimal policy at $i+1$ is to uniformly-sampled actions from the optimal actions of each dynamics model. If the optimal cumulative rewards are different, the action probabilities are weighted by the cumulative rewards of each dynamics model. On the other hand, partial trajectories from different dynamics models might predict the same $z$. If the optimal actions in the same state are conflicting, to increase the performance of objective Equation (1), the policy gradient has to backpropagate from $\pi$ to $\phi$. If the partial trajectories $\tau_{0:i}$ are different, the contexts in these partial trajectories would be distinctive. Finally, the output action distribution of the context-aware policy would be optimized in a subset of the dynamics models in which the partial trajectories $\tau_{0:i}$ are consistent.

## 4.3 Practical Implementation of Offline Model-based Adaptable Policy Learning

From the decision-making problem with Pak-DM to the real offline setting, the additional thing we should consider is the recognition of the inaccessible space and the construction of the dynamics model set. Especially in tasks with infinite state-action space, it is impractical to find the exact inaccessible space and recover all possible transitions in it. As a practical implementation, we use the ensemble technique to learn the dynamics model set, which would predict similar transitions in the accessible space and tend to predict different transitions in inaccessible space. With large ensemble models with different initialized weights, we can construct a large number of transition cases in the inaccessible space. If the real transition pattern falls into the variant of the ensemble dynamics model set, relying on the interpolation ability of the environment-context extractor $\phi$, the adaptable policy $\pi$ can take appropriate actions. However, only relying on the randomness of the initialization, to cover real cases in all out-of-support region need sufficiently enough dynamics model set, which would be highly expensive. In order to trade off the cost of model construction and better adaptivity, in the practical implementation, we use several tricks to constrain the policy exploration in the ensemble dynamics model set: 1) We add some mild constraints to the exploration region. To mitigate the compounding error of the model, we constrain the maximum rollout length to a fixed number. Besides, at each step, a penalty is calculated according to the model uncertainty $U(s_t, a_t)$, which measures the prediction uncertainty of the learned transition models at $(s_t, a_t)$. As a result, a reward provided to agents consists of two parts: a reward given by a reward function and a penalty calculated by $U(s_t, a_t)$. The constraints are the same as MOPO [12], but the coefficients are more relaxed; 2) As we increase the rollout length, the large compounding error might lead the agent to reach entirely unreal regions in which the states would never appear in the deployment environment. These samples are useless for adaptable policy learning and might make the training process unstable.

---

**Algorithm 1** Offline model-based adaptable policy learning

---

**Input**: $\phi_\varphi$ as an environment-context extractor, parameterized by $\varphi$; Adaptable policy network $\pi_\theta$ parameterized by $\theta$; Offline dataset $\mathcal{D}_{\text{off}}$; Trajectory termination rule $f$; Rollout horizon $H$;
**Process**:

    Generate an ensemble of $m$ dynamics models $\{\hat{\rho}_i\}$ via supervised learning
    Initialize an empty buffer $\mathcal{D}_{\text{rollout}}$; Add hidden states $z$ to each tuple in $\mathcal{D}_{\text{rollout}}$ and initialize with **0**
    **for** 1, 2, 3, ... **do**
        Randomly select the dynamics model $\hat{\rho}_i$ in $\{\hat{\rho}_i\}$ and sample $s_1, a_0, z_0$ from $\mathcal{D}_{\text{off}}$
        **for** $t$=1, 2 , ..., $H$ **do**
            Sample $z_t$ from $\phi_\varphi(z|s_t, a_{t-1}, z_{t-1})$ and then sample $a_t$ from $\pi_\theta(a|s_t, z_t)$
            Rollout one step $s_{t+1} \sim \hat{\rho}_i(s|s_t, a_t)$ and $r_{t+1} = r(s_t, a_t)$
            Compute the terminal state $d_{t+1} = f(s_{t+1})$
            Compute the reward penalty: $r_{t+1} \leftarrow r_{t+1} - \lambda U(s_t, a_t)$
            Add $(s_{t+1}, r_{t+1}, d_{t+1}, s_t, a_t, z_t)$ to $\mathcal{D}_{\text{rollout}}$
            Break the rollout if $d_{t+1}$ is True
        **end for**
        Update the stored hidden states $z$ in $\mathcal{D}_{\text{off}}$ with $\phi_\varphi$
        Use SAC [29] to update $\varphi$ and $\theta$ via Equation (1) with $\mathcal{D}_{\text{rollout}}$ and $\mathcal{D}_{\text{off}}$
    **end for**

---

Given a task, we can construct some simple rules to discriminate the entirely unreal regions and terminate the trajectory rollout (i.e., set the "done" flag to True) when reaching these regions. In our implementation, we terminate trajectories when the predicted next states are out of range of $(-s_{\min}, s_{\max})$, where $s_{\min}$ and $s_{\max}$ are two hyper-parameters to define a reasonable range of state space.

Based on the above techniques, we propose the practical implementation of offline model-based adaptable policy learning in Algorithm 1. More details can be found in the Appendix D.

## 5 Experiments

We evaluate MAPLE on multiple offline MuJoCo tasks [16]. All the details of MAPLE's training and evaluation are given in Appendix E and Appendix F. We release our code at Github [2].

### 5.1 Comparative Evaluation on Benchmark Tasks

Table 1: Results on MuJoCo tasks. Each number is the normalized score proposed by Fu et al. [30] of the policy at the last iteration of training, $\pm$ standard deviation. Among the offline RL methods, we bold the highest mean for each task.

| Environment | Dataset | MAPLE | MOPO | MOPO-loose | SAC | BEAR | BC | BRAC-v | CQL |
|---|---|---|---|---|---|---|---|---|---|
| Walker2d | random | **21.7 ± 0.3** | 13.6 ± 2.6 | 8.0 ± 5.4 | 4.1 | 6.7 | 9.8 | 0.5 | 7.0 |
| Walker2d | medium | 56.3 ± 10.6 | 11.8 ± 19.3 | 32.6 ± 18.0 | 0.9 | 33.2 | 6.6 | **81.3** | 79.2 |
| Walker2d | mixed | **76.7 ± 3.8** | 39.0 ± 9.6 | 35.7 ± 2.2 | 3.5 | 25.3 | 11.3 | 0.4 | 26.7 |
| Walker2d | med-expert | 73.8 ± 8.0 | 44.6 ± 12.9 | 66.7 ± 14.8 | -0.1 | 26.0 | 6.4 | 66.6 | **111.0** |
| HalfCheetah | random | **38.4 ± 1.3** | 35.4 ± 1.5 | 35.4 ± 2.1 | 30.5 | 25.5 | 2.1 | 28.1 | 35.4 |
| HalfCheetah | medium | **50.4 ± 1.9** | 42.3 ± 1.6 | 44.0 ± 1.6 | -4.3 | 38.6 | 36.1 | 45.5 | 44.4 |
| HalfCheetah | mixed | **59.0 ± 0.6** | 53.1 ± 2.0 | 36.9 ± 15.0 | -2.4 | 36.2 | 38.4 | 45.9 | 46.2 |
| HalfCheetah | med-expert | **63.5 ± 6.5** | 63.3 ± 38.0 | 15.0 ± 6.0 | 1.8 | 51.7 | 35.8 | 45.3 | 62.4 |
| Hopper | random | 10.6 ± 0.1 | 11.7 ± 0.4 | 10.6 ± 0.6 | 11.3 | 9.5 | 1.6 | **12.0** | 10.8 |
| Hopper | medium | 21.1 ± 1.2 | 28.0 ± 12.4 | 16.9 ± 2.4 | 0.8 | 47.6 | 29.0 | 32.3 | **58.0** |
| Hopper | mixed | **87.5 ± 10.8** | 67.5 ± 24.7 | 83.1 ± 6.5 | 1.9 | 10.8 | 11.8 | 0.9 | 48.6 |
| Hopper | med-expert | 42.5 ± 4.1 | 23.7 ± 6.0 | 25.1 ± 1.8 | 1.6 | 4.0 | 111.9 | 0.8 | **98.7** |

We test MAPLE in standard offline RL tasks with D4RL datasets [30]. The ensemble dynamics model set is trained via supervised learning. We repeat each task with three random seeds. In the

model learning stage, we train 20 models for each task and select 14 of them as the ensemble model for policy learning. The horizon $H$ is set to 10 in these tasks. The policy is trained for 1000 iterations in the policy learning stage.

We compare MAPLE with: (1) MOPO: Learn an ensemble model via supervised learning and learn a policy in the ensemble models with uncertainty penalty [12]; (2) MOPO-loose: MOPO algorithm with the same hyperparameter as MAPLE for constraints, which is looser than MOPO; (3) BEAR: Learn a policy via off-policy RL while constraining the maximum mean discrepancy of the current policy to the behavior policy [11]; (4) BC: Imitate the behavior policy via supervised learning; (5) SAC: Perform typical SAC updates with the static dataset [29]; (6) BRAC-v: A behavior-regularized actor-critic proposed by Wu et al. [21]; (7) CQL: Learn an action-value function with regularization to obtain a conservative policy [31]. Results of (3-7) and (1) are taken from the work of [30] and [12].

Table 1 has shown the performance of 12 tasks. In summary, the performance of MAPLE on 7 tasks is better than other SOTA algorithms. Besides, MAPLE reaches the best performance among the SOTA model-based conservative policy learning algorithms in **10 out of the 12 tasks**. These results demonstrate the superior generalization ability of MAPLE.

We can also find that the performance of MAPLE and MOPO are higher than other model-free baseline algorithms in most of the tasks, which reveals that model-based methods can find better policies by taking actions outside of the action distribution of the behavior policies. However, in Hopper experiments, model-free methods BC, CQL, and BRAC-v often reach better performance. We consider that it is because the environment is unstable, more diverse collected data is needed for robust dynamics model learning. Even in MAPLE, a finite number of ensemble dynamics models might not cover the real case so that to make a robust adaptation. Therefore, only in the "Hopper-mixed" task, which has diverse collected data, MOPO and MAPLE can improve the deployment performance.

We also conduct MOPO-loose for each task, which shares the same hyper-parameters with MAPLE, including the ensemble model size, rollout length, and penalty coefficient. The results show that for some cases (e.g., Walker2d-med-expert and Hopper-mixed), MOPO-loose can also enhance the performance. We consider the improvement coming from the constraints in original implementations to be too tight. However, in most of the tasks, the improvement is not as significant as MAPLE. This phenomenon indicates that the improvement of MAPLE does not come from parameter tuning but our self-adaptation mechanism.

The implementation of MAPLE shares the cross-domain idea with meta-RL, particularly domain randomization and sim2real. In the current implementation, the online system identification (OSI) method in meta RL is employed [26, 27, 28]. Other techniques, e.g., VariBAD [32], can also be integrated into MAPLE. We construct another variant of MAPLE, i.e., VariBAD-MAPLE, as another implementation. VariBAD-MAPLE can also reach a significantly better performance than MOPO. Compared with the original implementation of MAPLE, VariBAD-MAPLE can do better than MAPLE in Walker2d-medium and HalfCheetah-random but worse than MAPLE in Walker2d-mixed and HalfCheetah-mixed. The detailed results can be found in Appendix F.6.

## 5.2 Analysis of MAPLE

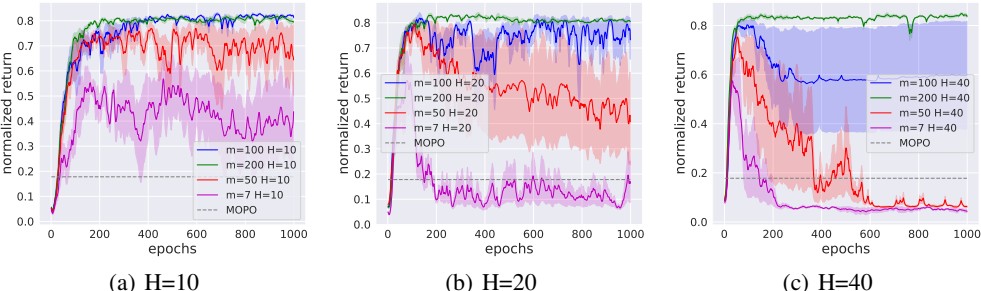

(a) H=10          (b) H=20          (c) H=40

Figure 3: The learning curves of MAPLE with different hyper-parameters $m$ and $H$. The solid curves are the mean of normalized return and the shadow is the standard error.

The ultimate target of MAPLE is handling the decision-making challenge in out-of-support regions. By constructing all possible transitions in out-of-support regions, the probing-reducing loop can find the optimal policy finally. However, it is impractical to construct a dynamics model set that covers all possible transitions in out-of-support regions. In practice, we construct an ensemble dynamics model set and use loose constraints on policy sampling to expand the exploration boundary for better asymptotic performance. Therefore, with a larger size of the model set, the dynamics models can cover more real transitions in out-of-support regions, then MAPLE is expected to have better performances by relaxing the constraints. In this section, to verify the argument, we analyze the relationship among constraint degree, the size of ensemble models, and the asymptotic performance.

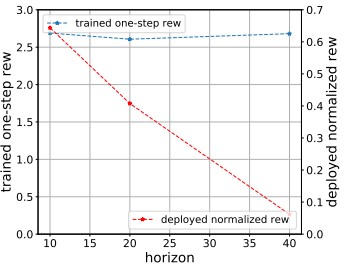

Figure 4: A Comparison of trained and deployed rewards at the 1000th epoch based on $m = 50$ and among different $H$.

In MAPLE, the constraint degree can be evaluated by the rollout length ($H$) or the reward penalty coefficient ($\lambda$); The size of ensemble models is $m$; The asymptotic performance is evaluated by the cumulative rewards at the 1000th iteration. We select Walker2d-medium to verify the argument. With the fixed $\lambda$, we compare the asymptotic performance of different $H$ and $m$.

We give the results in Figure 3. Firstly, for all $H$, by increasing the size of $m$, the performance of the converged policies is improved. On the other hand, without enough ensemble models, too loose constraints will result in worse performance. In particular, for the setting of $m = 50$, when $H = 10$, the final normalized return is near 0.7. However, as $H$ increases, the asymptotic performance drops gradually. When $H = 40$, the asymptotic performance is even worse than MOPO. As can be seen in Figure 4, the trained one-step reward is similar among different horizons. It means that the performance of policies in dynamics models is similar. The worse performance indicates that the adaptable policy overfits the finite dynamics models and $\phi$ fails to infer a correct environment-context to the adaptable policy when deployed. The issue can be remedied by constructing more dynamics models. As depicted in Figure 3(c), by increasing the model size to 200, the asymptotic performance recovers around 80%.

We also conducted additional experiments in Walker2d-mixed and HalfCheetah-mixed to visualize the $z$. The result reveals that different dynamics models are distinguished by $z$ and are approximated divided into several groups. Besides, we sample trajectories for 1000 time-step in the deployment environment. we found that the value of $Z$ oscillated within a region. The detailed results can be found in Appendix F.5.

### 5.3    MAPLE with large dynamics model set

Table 2: Results on MuJoCo tasks with MAPLE-200.

| Environment | Dataset | MAPLE-200 | MAPLE |
|---|---|---|---|
| Walker2d | random | **22.1 ± 0.1** | 21.7 ± 0.3 |
| Walker2d | medium | **81.3 ± 0.1** | 56.3 ± 10.6 |
| Walker2d | mixed | 75.4 ± 0.9 | **76.7 ± 3.8** |
| Walker2d | med-expert | **107.0 ± 0.8** | 73.8 ± 8.0 |
| HalfCheetah | random | **41.5 ± 3.6** | 38.4 ± 1.3 |
| HalfCheetah | medium | 48.5 ± 1.4 | **50.4 ± 1.9** |
| HalfCheetah | mixed | **69.5 ± 0.2** | 59.0 ± 0.6 |
| HalfCheetah | med-expert | 55.4 ± 3.2 | **63.5 ± 6.5** |
| Hopper | random | **10.7 ± 0.2** | 10.6 ± 0.1 |
| Hopper | medium | **44.1 ± 2.6** | 21.1 ± 1.2 |
| Hopper | mixed | 85.0 ± 1.0 | **87.5 ± 10.8** |
| Hopper | med-expert | **95.3 ± 7.3** | 42.5 ± 4.1 |

Based on the above analysis, we can get an empirical conclusion that *increasing the model size is significantly helpful to find a better and robust adaptable policy via expanding the exploration boundary*. Therefore, we conduct another variant of the MAPLE algorithm, MAPLE-200, which uses 200 ensemble dynamics models for policy learning and expands the rollout horizon to 20.

The results of MAPLE-200 can be found in Table 2. We can see that the empirical conclusions not only suit the Walker2d-medium but also other tasks. In all of the tasks, MAPLE-200 reaches at least similar performance to MAPLE. In the tasks like Walker2d-med-expert, HalfCheetah-mixed, Hopper-medium, and Hopper-med-expert, the performance improvement of MAPLE-200 is significant. Besides, in the tasks of

Hopper-medium and Hopper-med-expert, where MAPLE and MOPO fail to reach a comparable performance to model-free offline methods, MAPLE-200 can reach similar or much better results.

MAPLE-200 demonstrates a powerful adaptation ability. However, we point out that, by increasing the 10x size of ensemble dynamics models, the time overhead for MAPLE-200 training is also larger. For example, by using NVIDIA Tesla P40 and Xeon(R) E5-2630 to train the algorithms, the time overhead of MAPLE-200 is 10 times longer than MAPLE. Besides, to obtain dynamics models that covered more real cases in out-of-support regions than MAPLE-200, the size of ensemble models becomes much larger, which is one of the limitations for current MAPLE implementation.

## 6    Discussion and Future Work

Prior work has demonstrated the feasibility of model-based policy learning in the offline setting by using the conservative policy learning paradigm [12, 15]. It is an important breakthrough from zero to one in the offline setting, but it is far from the end of offline model-based policy learning. In this work, we investigate the decision-making problems in out-of-support regions directly. We first formulate the problem as decision-making in Pak-DM and propose MAPLE, a learn-to-adapt paradigm to solve the problem. We also give a theorem to describe the pros and cons of the paradigms to give us principles for the paradigm selection. We verified MAPLE in the MuJoCo tasks, and get our empirical conclusion: by increasing the size of the model set, we can expand the exploration boundary in the approximated dynamics models by using adaptable policy to make better and robust decisions in deployment environments.

MAPLE gives another direction to handle the offline model-based learning problem: Besides constraining on sampling and training dynamics models with better generalization, we can model out-of-distribution regions by constructing all possible transition patterns. The current limitation lies in the implementation: (1) The extractor's generalization ability depends on the neural network itself, which is uncontrollable to some degree. Adding some auxiliary tasks might handle this issue; (2) Ensemble is the direct way to cover real transitions in out-of-support regions. However, as the size of the model set becomes large, it is hard to generate new different dynamics models to cover the real cases only by increasing the size of the model. More efficient ways to increase the cover real transitions of the dynamics model set should be further explored.

## Acknowledgements

This work is supported by the National Key Research and Development Program of China (2020AAA0107200) and the NSFC (61876077). We thank Zheng-Mao Zhu, Shengyi Jiang, Rong-Jun Qin, Yu-Ren Liu, and the anonymous reviewers for their useful suggestions on the papers.

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
