# A  Proof

**Theorem 1** *Given a target dynamics model $\rho$, a policy $\pi_a$ learned by adapting, a policy $\pi_c$ learned by constraints, and the maximum step $N_m$ taken by $\pi_a$ to probe and reduce the policy set to a single policy, the performance gap between $\pi_a$ and $\pi_c$ satisfies:*

$$J_\rho(\pi_a) - J_\rho(\pi_c) \geq \Delta_c - \Delta_p - \gamma^{N_m+1} J_{\rho_\Delta}(\pi^*),$$

*where $\Delta_c$ denotes the performance gap of the optimal policy $\pi^*$ and $\pi_c$, while $\Delta_p$ denotes the performance degradation of MAPLE compared with $\pi^*$ because of the phase of probing. $J_{\rho_\Delta}(\pi^*)$ denotes the performance degradation of $\pi^*$ on the dynamics model $\rho$ caused by different initial state distribution: $J_{\rho_\Delta}(\pi^*) = \mathbb{E}_{d^{\pi^*}_{N_m+1}(s)}[V^\star(s)] - \mathbb{E}_{d^{\pi_a}_{N_m+1}(s)}[V^\star(s)]$, where $d^{\pi^*}_{N_m+1}(s)$ and $d^{\pi_a}_{N_m+1}(s)$ denote the state distribution induced by $\pi^*$ and $\pi_a$ at the $N_m+1$ step and $V^*(s)$ denotes the expected long-term rewards of $\pi^*$ at state $s$.*

**Proof A.1** *MAPLE algorithm includes two policies: a probing policies $\pi_p$ to visit state-action pair in inaccessible space, and the reduced policy $\pi_t$, which is equal to the optimal $\pi^*$ in theory. Given a trajectory, the cumulative reward is:*

$$\sum_{k=0}^{N} \gamma^k r_k^p + \sum_{k=N+1}^{T} \gamma^k r_k^*,$$

*where $N$ denotes the time-step for the probing policy $\pi_p$ to reduce the policy set to the reduced policy $\pi_t$, which is equal to $\pi^*$. $r^p$ and $r^*$ denote the rewards run by $\pi^p$ and $\pi_t$ respectively. Regarding the policy of MAPLE as a mixed policy $\pi_a$, the performance can be written as:*

$$J(\pi_a) = \mathbb{E}_{\tau \sim p(\tau|\pi_a,\rho)} \left[ \sum_{k=0}^{N(\tau)} \gamma^k r_k^p + \sum_{k=N(\tau)+1}^{T} \gamma^k r_k^* \right],$$

*where $N(\tau)$ denotes a function that outputs the time-step needed for the probing policy $\pi_p$ to reduce the policy set for each trajectory $\tau$. Assuming the maximized time-step needed is $N_m$, that is $N_m := \max_\tau N(\tau)$, we have:*

$$
\begin{aligned}
J(\pi_a) \geq & \mathbb{E}_{\tau \sim p(\tau|\pi_a,\rho)} \left[ \sum_{k=0}^{N_m} \gamma^k r_k^p + \sum_{k=N_m+1}^{T} \gamma^k r_k^* \right] \\
= & \int \rho_0(s) \mathbb{E}_{s_0=s,\tau \sim p(\tau|\pi_p,\rho)} \left[ \sum_{k=0}^{N_m} \gamma^k r_k^p \right] ds \\
& + \gamma^{N_m+1} \int \rho_p(s) \mathbb{E}_{s_{N_m+1}=s,\tau \sim p(\tau|\pi^*,\rho)} \left[ \sum_{k=0}^{T-N_m-1} \gamma^k r_{k+N_m+1}^* \right] ds \\
= & J_\rho^{\mathrm{par}}(\rho_0,\pi^P,N_m) + \gamma^{N_m+1} J_\rho^{\mathrm{par}}(\rho_p,\pi^*,T-N_m-1),
\end{aligned}
$$

*where $\rho_0(s)$ denotes the initial state distribution of the environments, and $\rho_p(s)$ denotes the state distribution at time-step $N_m$ running with $\pi_p$. Here we introduce a new notation $J_\rho^{\mathrm{par}}(\rho_0,\pi,T)$ to describe the partial performance of policy $\pi$ in dynamics model $\rho$, running with the initial state distribution $\rho_0$ and the horizon $T$. Then we know the performance gap between $\pi^*$ and $\pi_a$:*

$$
\begin{aligned}
J(\pi^*) - J(\pi_a) \leq & J_\rho^{\mathrm{par}}(\rho_0,\pi^*,N_m) - J_\rho^{\mathrm{par}}(\rho_0,\pi^P,N_m) \\
& + \gamma^{N_m+1} J_\rho^{\mathrm{par}}(\rho^* - \rho_p,\pi^*,T-N_m-1),
\end{aligned}
$$

*where $\rho^*(s)$ denote the state distribution at time-step $N_m$ running with $\pi^*$. Assuming the performance gap between $\pi^*$ and $\pi_c$ is :*

$$J(\pi^*) - J(\pi_c) = \Delta_c,$$

*we have:*

$$J(\pi_a) - J(\pi_c) \geq \Delta_c - \left(J_\rho^{\mathrm{par}}(\rho_0, \pi^*, N_m) - J_\rho^{\mathrm{par}}(\rho_0, \pi^p, N_m)\right)$$
$$- \gamma^{N_m+1} J_\rho^{\mathrm{par}}(\rho^* - \rho_p, \pi^p, T - N_m - 1).$$

$J_\rho^{\mathrm{par}}(\rho^* - \rho_p, \pi^*, T - N_m - 1)$ *is the performance degradation of* $\pi^*$ *on the dynamics model* $\rho$ *caused by different initial state distributions. In the infinite horizon setting, given an initial state* $s$, $\sum_{k=0}^{T-N_m-1} \gamma^k r_{k+N_m+1}^* = V^*(s)$, *where* $V^*(s)$ *denotes the expected long-term rewards of* $\pi^*$ *taking state* $s$ *as the initial state. Letting* $d_{N_m+1}^{\pi^\star}(s)$ *and* $d_{N_m+1}^{\pi_a}(s)$ *denote the state distribution induced by* $\pi^\star$ *and* $\pi_a$ *at the* $N_m + 1$ *step, we have:*

$$J_\rho^{\mathrm{par}}(\rho^* - \rho_p, \pi^*, T - N_m - 1) = \mathbb{E}_{d_{N_m+1}^{\pi^\star}(s)}[V^\star(s)] - \mathbb{E}_{d_{N_m+1}^{\pi_a}(s)}[V^\star(s)].$$

*Letting* $\Delta_p := J_\rho^{\mathrm{par}}(\rho_0, \pi^*, N_m) - J_\rho^{\mathrm{par}}(\rho_0, \pi^p, N_m)$, *which is the performance degradation of MAPLE compared with* $\pi^*$ *in the phase of probing in inaccessible space, and* $J_{\rho_\Delta}(\pi^*) := J_\rho^{\mathrm{par}}(\rho_\Delta, \pi^*, T - N_m - 1)$, *which is the performance of* $\pi^*$ *on the dynamics model* $\rho$ *with an occupancy measure gap of the state distribution* $\rho_\Delta = \rho^* - \rho_p$ *at time-step* $N_m$, *we get our conclusion:*

$$J_\rho(\pi_a) - J_\rho(\pi_c) \geq \Delta_c - \Delta_p - \gamma^{N_m+1} J_{\rho_\Delta}(\pi^*).$$

# B  Decision-Making in Out-of-Support Regions: A Toy-Environment Verification

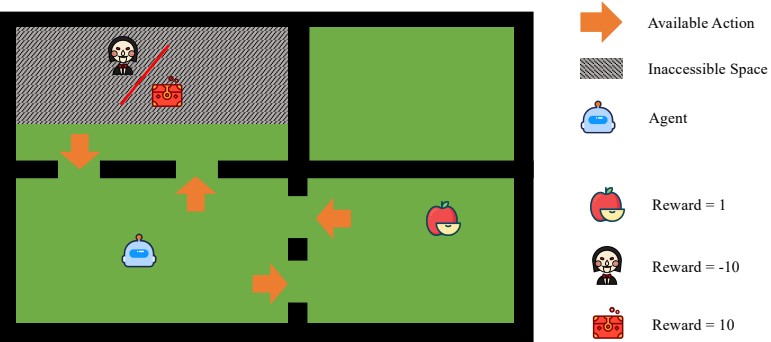

Figure 1: Illustration of the Treasure-Explorer task. In this task, the policy needs to control an agent to move in the map. There are three reachable rooms in the environment (i.e., the lower-left room, the lower-right room, and the upper-left room). In each room, the agent can choose one of the available actions in the room (marked as the yellow arrows). By taking one of the available actions, the agent will move to the room pointed by the arrow. After moving into different rooms, the agent will hit the object in the room and get a reward, then get back to the lower-left room immediately. This is an environment with uncertainty. In the upper-left room, the object in the inaccessible space is unknown. That is, we do not know the next state by taking the "up" action in the lower-left room. The inaccessible space here is simplified by omitting the possibility of reaching the other two rooms to simplify the analysis.

In this work, we give a probe-reduce paradigm for decision-making in out-of-support regions. In Section 4.2, we use an environment-context extractor and an adaptable policy to build the practical algorithm. We claim that with the RNN-based environment-context extractor $\phi$ optimized with Equation (1), the context-aware policy $\pi$ can automatically probe environments and reduce the policy set. In this section, we design a toy environment, Treasure Explorer, to verify the argument. Figure 1 illustrates the Treasure-Explorer environment. Treasure Explorer follows the problem of decision-making with a partially known dynamics model (Pak-DM) in a surrogate objective. We learn an adaptable policy and a conservative policy in the environment and show the decision process of the adaptable policy matching the claims in the previous sections.

## B.1 Experimental Setting

In this environment, we only have a partially known dynamics model as illustrated in Figure 1 and would like to maximize the reward in a deployed environment. The deployed environment could be one of the three dynamics models: ghost, nothing or treasure, as illustrated in Figure 2(c). In the task, the agent is initialized at the lower-left room. By taking one of the available actions, the agent will move to the room pointed by the arrow. After moving into different rooms, the agent will hit the object in the room and get a reward, then get back to the lower-left room immediately. We set $r = 1$ when hitting the apple, $r = 10$ by hitting the treasure, and $r = -1$ by hitting the ghost. The agent can make decisions 5 times. That is, the horizon of a trajectory is set to 5.

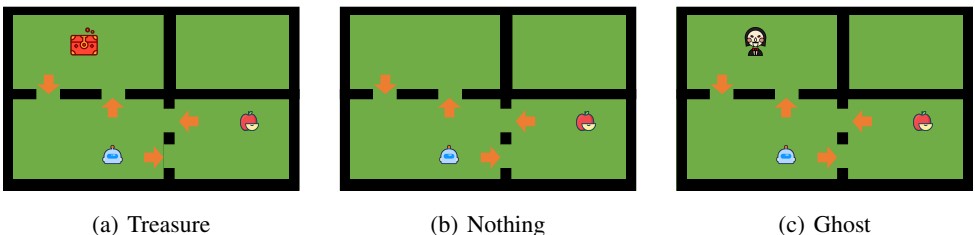

| (a) Treasure | (b) Nothing | (c) Ghost |

Figure 2: All possible dynamics models.

In the following, we give the optimal policy and the ideal probe-reduce policy:

**Optimal policy**    The optimal policy depends on the object in the inaccessible space. If the object in the inaccessible space gives -10 (ghost) or 0 (nothing) reward, the optimal policy is going right every time. If the object provided 10 reward (treasure), the optimal policy is going up forever.

**Probe-reduce policy**    Ideally, a probe-reduce policy will first probe the inaccessible space: it will choose to go up and observe the object in the upper-left room. Based on the object obtained in the upper-left room, the possible dynamics model set will be reduced to one of the three models. Subsequently, it will choose the optimal policy to control the robot according to the confirmed dynamics model.

## B.2 Training

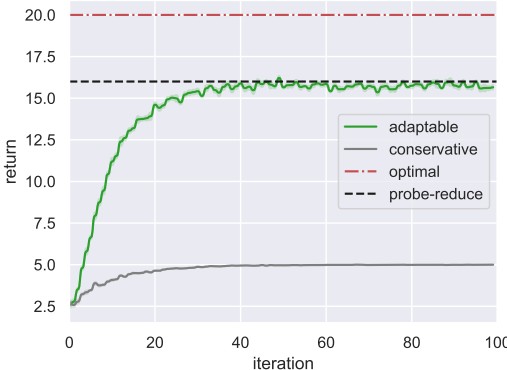

Figure 3: The learning curves of the adaptable policy and conservative policy. We evaluate the return by averaging the cumulative rewards of trajectories that are run in randomly selected environments from the dynamics model set. The solid curves are the mean returns, which are shadowed by the standard errors of ten random seeds.

We learn an adaptable policy and a conservative policy in the Treasure-Explorer environment. For the adaptable policy learning, we randomly select dynamics models from the dynamic model set with the

same probability for each time of resetting the environment. For the conservative policy learning, we learn a policy in the Treasure-Explorer environment but with large penalties when the agent reaches the inaccessible space. We instantiate the conservative policy with MLP. In contrast, we construct the environment-context extractor with GRU and the adaptable policy with MLP. We adopt PPO to optimize the policies.

## B.3 Training Performance

Figure 3 shows the learning curve of each method. There are two dash lines indicating the theoretical performance of the optimal policy and the probe-reduce policy. We can see that the performance of the adaptable policy can converge to the theoretical performance of the probe-reduce policy. In terms of the performance in the dynamics model set, the results demonstrate that the learned adaptable policy is consistent with the probe-reduce policy. The performance gap of the conservative policy and the adaptation ability in the training environments depend on the sampling strategy of the adaptable policy learning. With a larger probability of training with the treasure dynamics model, the return of adaptable policy would be larger.

## B.4 Policy Behaviours in the deployed environment

In order to investigate the behavior patterns of the policies in different deployed environments, we record the up-to-now cumulative rewards at each step in a single trajectory. In particular, for the $i$-step, the up-to-now cumulative rewards is $\sum_{k=0}^{i} r_k$. The result is presented in Figure 4.

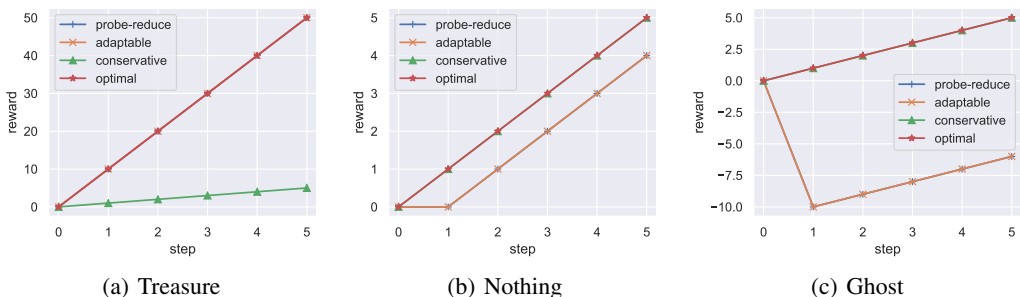

|  (a) Treasure | (b) Nothing | (c) Ghost |

Figure 4: The up-to-now cumulative rewards at each step for adaptable, conservative, probe-reduce, and optimal policies.

As can be seen in Figure 4, since the adaptable policy tries to probe the environment at the first step, the first-step reward is varied to the environments. After that, the RNN recognizes the environment, and the agent adjusts its policy to the optimal policy for the environment. As can be seen in Figure 4, after the first step, the slope of the up-to-now cumulative rewards of the adaptable policy are the same to the optimal policy. In contrast, the conservative agent always chooses to exploit the room with 1-reward. Besides, the cumulative rewards curve of the probe-reduce policy is the same as the adaptable policy.

We also present the trajectory sampled by the optimal policy, conservative policy, and adaptable policy in Figure 5. In the Nothing and the Ghost environment, the adaptable policy takes an active probing behavior at the first step. Then, it repeats the optimal policy for the rest steps. However, the conservative policy keeps a single behavior pattern: always picks the apple in the room on the right side. The trajectories match the behaviors of the probe-reduce policy: probing first and then exploiting accordingly.

This result supports the claim in Section 4.2, the adaptable policies indeed possess such a probing-reduce characteristic. Without such adaptability, the agent is prone to stick at a local minimum solution and produces a conservative policy.

The results are also consistent with the Theorem 1. As can be found in the case of "Ghost", tasks with large penalties on undesirable behavior might make $\Delta_p$ larger, which reduces the overall performance of the adaptable policy. In this case, the performance of the adaptable policy is significantly worse

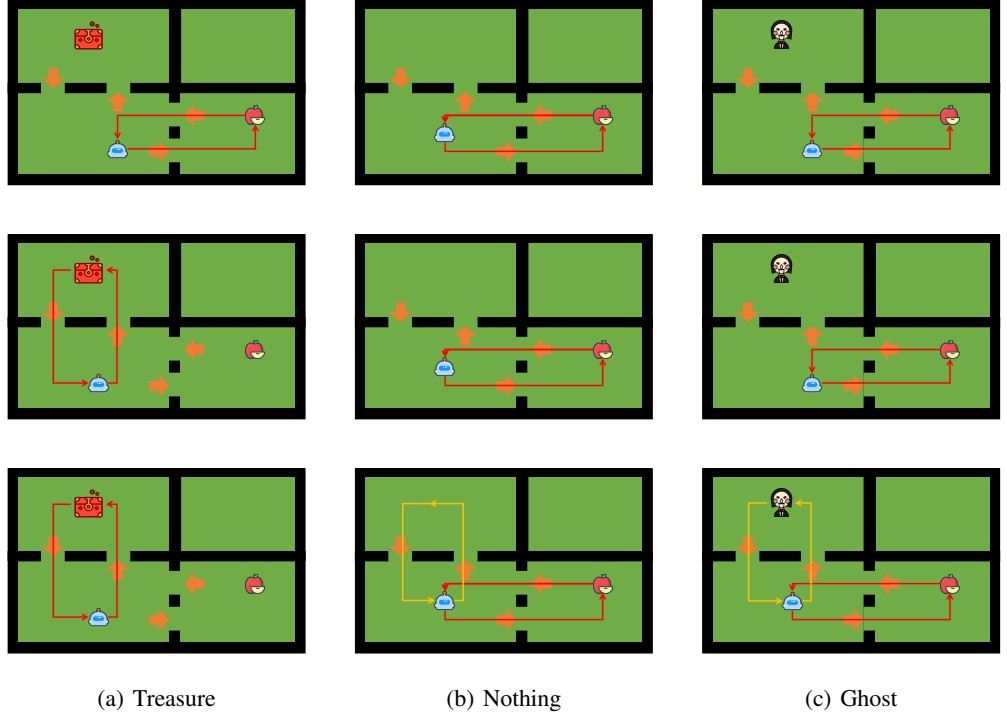

(a) Treasure          (b) Nothing          (c) Ghost

Figure 5: Trajectories of the conservative, adaptable, and optimal policies in each environment. The first row is the conservative policy; the second is the optimal policy; the third is the adaptable policy. In the Nothing and the Ghost environments, a yellow line indicates the probing behavior. The agent will execute the probing behavior for one time, and then continuously perform the red line behavior. In the Treasure environment, the probing behavior is the same as the optimal behavior.

than the conservative policy. On the other hand, as in the case of "Treasure", the adaptable policy can reach a better performance than the conservative policy, with a larger performance gap of $\Delta_c$.

## C  Additional Related Work

MAPLE uses a meta-learning technique of policy adaptation to design the practical algorithm of decision-making in out-of-support regions. Meta-learning [1] provides a way for policy adaptation. In meta-learning, a meta-policy model [2, 1] is learned from a set of source tasks and will adapt for a new environment with a small number of samples. Meta-based methods often need additional trajectories in the target environment to update its parameter. Online system identification (OSI) methods is to identify the environments automatically [3, 4, 5, 6, 7]. OSI methods always contain a universe policy (UP) and an environment-context extractor. The environment-context extractor extracts the information of the environment [4, 3, 6, 7]. The UP infers the action with the concatenation of the environment information and state as input. An environment-context extractor identifies the environment as the policy executing and does not need to update its parameter like meta-based methods. Thus, OSI techniques possess the ability to adapt to a new environment online.

## D  MAPLE in Implementation

### D.1  Network Structure

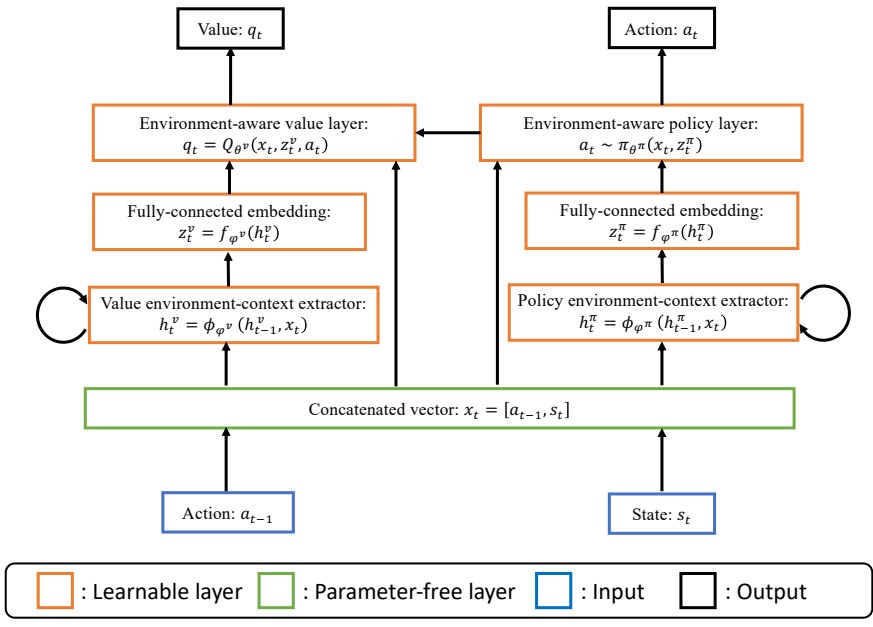

Figure 6: Illustration of the network structure for MAPLE.

The network structure of MAPLE is shown in Figure 6. In the implementation, we use two independent neural network parameters for the Q-value function and policy. In Figure 6, we use the superscript of $v$ and $\pi$ for the parameters $\varphi$ and $\phi$ to denote the independent neural network parameters for the Q-value function and policy. In MAPLE, there is a skip-connection for the environment-aware layer to reuse the original input features. The environment-context extractor layers are both modeled with a single-layer GRU cell [8]. Here we use $h_t^v$ and $h_t^\pi$ to denote the output of the two environment-context extractors. After the environment-context extractor layers, we use another fully-connected embedding layer $f$ to reduce the output of the hidden context. Empirically, the output of the fully-connected embedding $z$ should not be much larger than the dimension of $x_t$. Too large of $z$ might lead to unstable policy and Q-value training. The embedding layer and environment-aware layer are modeled with Multilayer Perceptron (MLP). Table 1 reports the detailed parameters of the neural network. The hyper-parameters have not been fine-tuned, other structures can be tried: e.g., we found that the performance would be better if increasing the fully-connected embedding layers to 128 at least in HalfCheetah tasks.

Table 1: The hyper-parameters of MAPLE for the neural network.

| Hyperparameter | Value |
|---|---|
| Activation function of hidden layers | relu |
| Activation function of policy output | tanh |
| Activation function of q-value output | linear |
| Fully-connected embedding layers $f$ | [16] |
| Unit of the environment-context extractor $\phi$ | 128 |
| Environment-aware layers of $Q$ and $\pi$ | [256, 256] |

### D.2 Implementation details

**Reward penalty with uncertainty quantification:** We use reward penalty and trajectory trunca-tion as in MOPO [9], but the coefficients are more relaxed (i.e., smaller reward penalty coefficient and longer rollout length). We model the dynamics models with Gaussian distribution. For each time-step $t$, the reward penalty $U(s_t, a_t) = \max_i ||\Sigma^i(s_t, a_t)||_2$, where $\Sigma^i(s_t, a_t)$ denotes the standard deviation of the predicted Gaussian distribution of the $i$-th dynamics model at state $s_t$ and action $a_t$, and $|| \cdot ||_2$ denotes the l2-norm.

**State/penalty clipping:** As we increase the rollout length, the large compounding error might lead the agent to reach entirely unreal regions in which the states would never appear in the deployment environment. These samples are useless for adaptable policy learning and might make the training process unstable. Thus, we constrain the range of state to [-100, 100], and we also truncate trajectories when the predicted next states are out of range of [-100, 100]. For the same reason, the reward penalty is clipped to [0, 40].

**Hidden state of RNN:** As seen in Algorithm 1, when sampling initial states for the model-policy interaction, the hidden states are also sampled. The policy will also start inferring action starting from the hidden state. Thus, we should update the hidden states of the offline dataset $\mathcal{D}_{off}$ periodically. In our implementation, for every four epoch, we infer the hidden states of policy and value function for the dataset.

### D.3 Hyper-parameters

Other hyper-parameters include: rollout length $H$, dynamics model size $m$, and penalty coefficient $\lambda$. For all of the tasks, we use $H = 10$, $m = 20$ and penalty coefficient $\lambda = 0.25$ except $\lambda = 4.0$ in HalfCheetah-med-expert. The other hyper-parameters are the same as the original MOPO.

## E Detailed Experimental Setting in MuJoCo tasks

We test the algorithms in standard offline RL tasks with D4RL datasets [10]. In particular, we use data from 3 environments: Hopper, HalfCheetah, and Walker2d. In each environment, we test MAPLE with 4 four kinds of datasets: random, medium, mixed, and medium expert (med-expert). The datasets are gathered through different strategies: random and medium are collected by a random and medium policy, while med-expert is the concatenation of medium and the data collected by an expert policy. Mixed uses the replay buffer of a policy trained up to the performance of the medium agent. We repeat each task with three random seeds. In the model learning stage, we train 20 models for each task and select 14 of them as the ensemble model for policy learning. The ensemble dynamics model set is trained via supervised learning. The policy is trained for 1000 iterations.

## F Extra Experimental Results in MuJoCo Tasks

### F.1 Learning Curves of MAPLE and MOPO-loose

The learning curves of MAPLE and MOPO-loose in 12 tasks can be found in Figure 11. MOPO-loose shares the same hyper-parameters with MAPLE, including the ensemble model size, the weights of

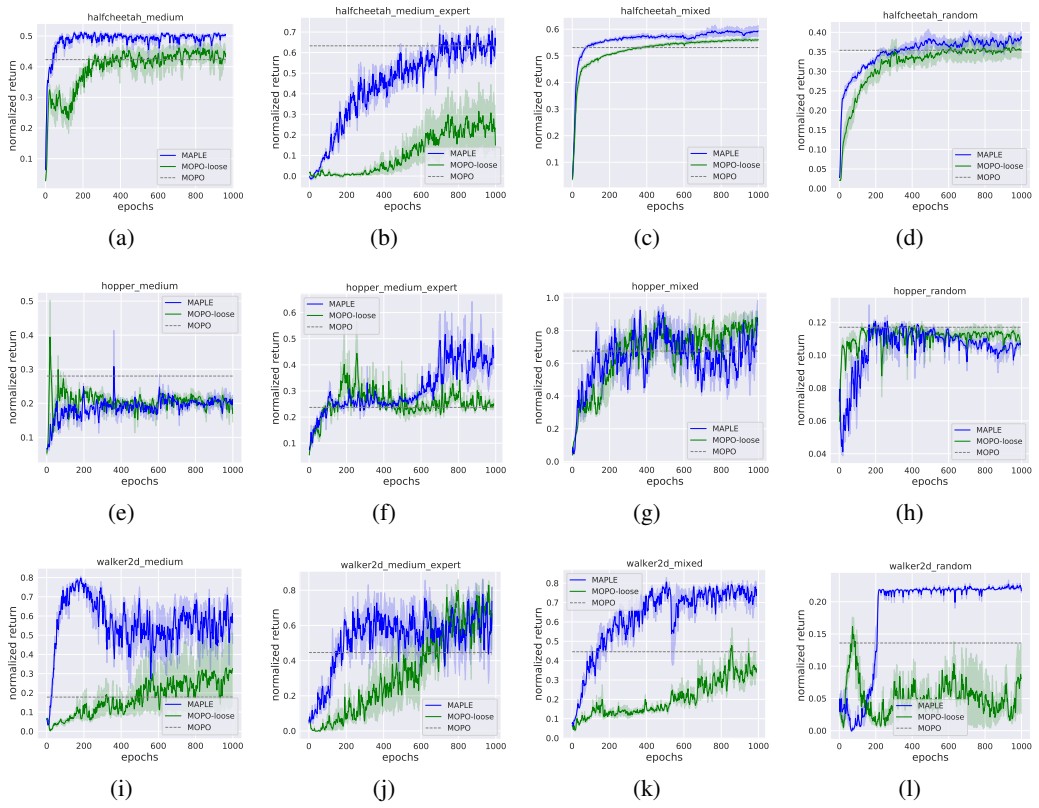

Figure 7: The learning curves of MAPLE, MOPO, and MOPO-loose in mujoco tasks. The solid curves are the mean reward and the shadow is the standard error of three seeds.

parameters of each dynamics model, rollout length, and penalty coefficient. The results show that for some cases (e.g., Walker2d-med-expert and Hopper-mixed), MOPO-loose can also enhance the performance. We consider the improvement coming from the constraints in original implementations being too tight. However, in most of the tasks, the improvement is not as significant as MAPLE. This phenomenon indicates that the improvement of MAPLE does not come from parameter tuning but our self-adaptation mechanism.

## F.2 Hyper-parameters Analysis

We test the performance of MAPLE with different rollout length $H$, dynamics model size $m$, and penalty coefficient $\lambda$. We search over $H \in \{5, 10, 40\}$, $m \in \{7, 20, 50\}$ and $\lambda \in \{1.0, 0.25, 0.05\}$ in the task Walker-medium, Halfcheetah-mixed, and Hopper-medium-expert. For each setting, we run with three random seeds. We summarize the results in Table 3 and Figure 8. We give the empirical conclusions as follows:

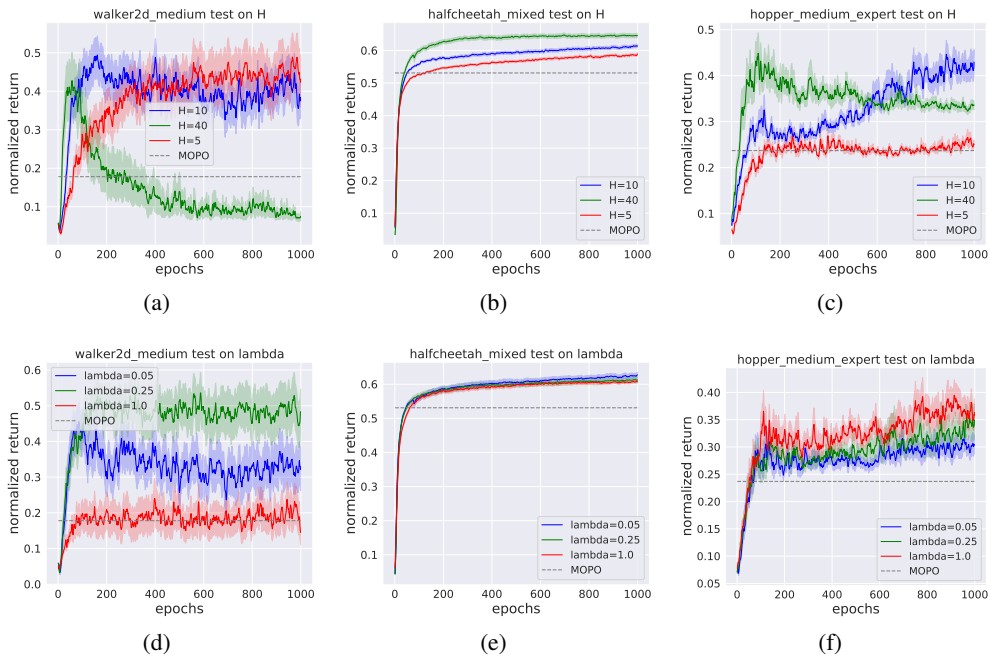

Figure 8: Illustration of hyper-parameters analysis. In Figure 8(a), Figure 8(b) and Figure 8(c), we group the results by $H$. For each group, there have nine different experiments. Similarly, in Figure 8(d), Figure 8(e) and Figure 8(f), we group the results by $m$.

First, as claimed in this work, tight constraints indeed lead to conservative policies, but the threshold starting to suppress the asymptotic performance of policies is varied to the tasks. As can be seen in Figure 8(d), in the tasks of walker-medium, $\lambda = 1.0$ is the setting significantly suppressing the asymptotic performance, while in hopper-medium-expert task, as can be seen in Figure 8(f), $\lambda = 1.0$ is a good setting for policy learning. In hopper-medium-expert and halfcheetah-mixed tasks, $H = 5$ is the setting significantly suppressing the asymptotic performance (see Figure 8(c) and Figure 8(b)), while $H = 5$ and $H = 10$ are both good settings in walker-medium task (see Figure 8(b) and Figure 8(a)).

Second, as we do not construct all possible transitions in out-of-support regions, too loose constraints also hurt the asymptotic performance. In the task of walker2d-medium, $H = 40$ makes the asymptotic performance much worse than other settings. In the task of hopper-medium-expert, $\lambda = 0.05$ and $H = 40$ also degenerate the asymptotic performance. However, in the task of halfcheetah-mixed, the performance can always be improved by relaxing the constraints. The reason is probably that the environment of halfcheetah is more stable than other tasks. In the halfcheetah environment, trajectories will not pre-terminate no matter how badly the agent behaves. Therefore, the environment is tolerable to some incorrect actions. While in the rest tasks, the agent might reach unsafe states after performing some undesired actions. The incorrect actions might come from the probing process of MAPLE and errors of inference of the environment-context for some time-steps. Since the environment is tolerable to some incorrect actions, the probing processing and the generalization error of the environment-context extractor have less impact on the deployment performance.

Third, for any type of constraint, increasing the size of the dynamics model set can improve the performance of MAPLE. Besides, as increasing enough size of the dynamics model set, the setting

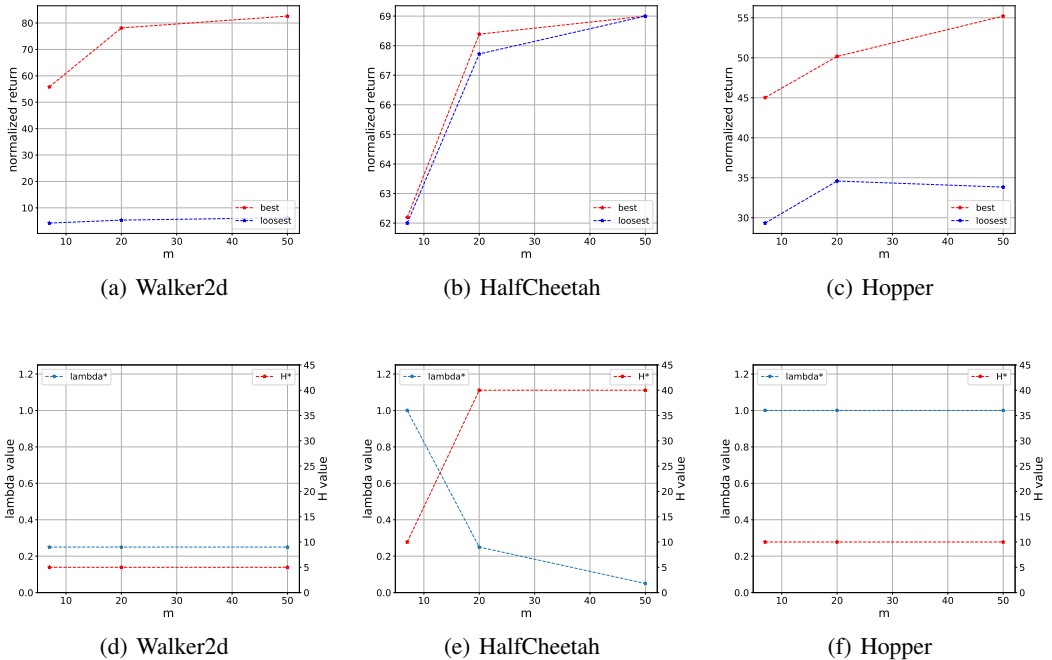

(a) Walker2d      (b) HalfCheetah      (c) Hopper

(d) Walker2d      (e) HalfCheetah      (f) Hopper

Figure 9: Illustration of hyper-parameters analysis on $m$. In the first row, we compare the normalized return of the best setting and the loosest setting. The x-axis is the model size $m$. For each $m$, the legend "best" is the setting that has the largest performance, among which model size is $m$. The legend "loosest" is the setting that $\lambda = 0.05$ and $H = 40$. In the second row, we compare the best constraint setting for each model size $m$. For each $m$, the legend "lambda*" is the setting that $\lambda$ value of the best-performance setting among which model size is $m$. Similarly, the legend "H*" is the setting that $H$ value of the best-performance setting among which model size is $m$.

with loose constraints can reach better performance. The illustration can be found in Figure 9. As can be seen in Figure 9(a), Figure 9(b) and Figure 9(c), as $m$ increased, the best-setting performances are significantly improved. Besides, as can be seen in Figure 9(e) and Figure 9(b), in the task of halfcheetah-mixed, when $m$ increased, the best-setting is relaxed (i.e., $\lambda*$ is reduced and $H*$ is longer) and the performance of the loosest setting is close to the best setting gradually. However, in the task of hopper-medium-expert and walker-medium, the performances of the loosest settings are significantly worse than the best setting. The constraints of $H*$ and $\lambda*$ in the best setting keep the same from $m = 7$ to $m = 50$. We think the reason is that the size of dynamics models is not so large enough to cover the real cases in the out-of-support regions within the explorable boundary. Thus, the extractor $\phi$ can not infer a correct environment-context when deployed.

The learning curves of each setting can be found in Figure 14, Figure 15 and Figure 16.

### F.3   Hyper-parameters Analysis with a larger size of dynamics models

In Appendix F.2, we found that the setting with the loosest constraints can not reach better performance as $m$ increased in Hopper-medium-expert and Walker2d-medium, and we think the reason is that the size of the dynamics model is not large enough. In this section, we select $\lambda = 0.25$ for Walker2d-medium and $\lambda = 1.0$ for Hopper-medium-expert, and compare the relation between $H$ and $m$. We search over $H \in \{10, 20, 40\}$, $m \in \{7, 50, 100, 200\}$. The results can be found in Figure 17.

We merge the results in Appendix F.2 and give an illustration of the relation between $H$ and $m$ in Figure 17. We can see that, by increasing $m$, the best setting is relaxed (i.e., $H*$ is longer) and the performance of the loosest setting is close to the best setting gradually. As can be seen in Figure 17, in Walker2d-medium, when $m = 200$, from $H = 10$ to $H = 40$, the normalized returns are all around $0.8$, which also shows the robustness of MAPLE to different constraints with large model size.

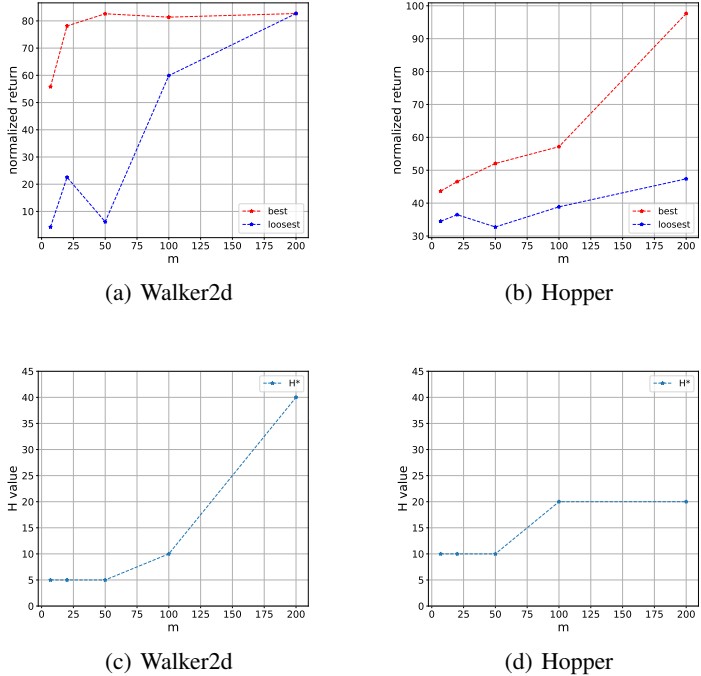

(a) Walker2d

(b) Hopper

(c) Walker2d

(d) Hopper

Figure 10: Illustration of hyper-parameters analysis on $m$. In the first row, we compare the normalized return of the best setting and the loosest setting. The x-axis is the model size $m$. For each $m$, the legend "best" is the setting that has the largest performance, among which model size is $m$. The legend "loosest" is the setting that $H = 40$. In the second row, we compare the best constraint setting for each model size $m$. For each $m$, the legend "H*" is the setting that $H$ value of the best-performance setting among which model size is $m$.

In Hopper-medium-expert, although there is still a gap between the performance of the best setting and the loosest setting, the performance of the loosest setting is gradually increased as $m$ increased and the $H*$ is relaxed from 10 to 20.

### F.4 Learning Curves of MAPLE-200

Based on the above analysis, we can get an empirical conclusion that increasing the model size is significantly helpful to find a better and robust adaptable policy via expanding the exploration boundary. Therefore, we conduct another variant of the MAPLE algorithm, MAPLE-200, which uses 200 ensemble dynamics models for policy learning and expands the rollout horizon to 20. We select $\lambda = 0.25$ for tasks in HalfCheetah and Walker2d, and $\lambda = 1.0$ for Hopper. we still select $\lambda = 4.0$ and $H = 10$ in the task of HalfCheetah-med-expert. We test MAPLE-200 in all of the tasks. The results can be found in Figure 18.

### F.5 Virtualization of the Hidden State $z$

We conduct additional experiments in Walker2d-mixed and HalfCheetah-mixed to visualize the $z$. We first randomly select $10,000$ states from the offline dataset and rollout the adaptable policy for 10 steps with each of the dynamics models in the ensemble model set. We record the absolute value of the context-variable $y_i = \frac{|z|_i}{|\mathrm{Dim}(\mathcal{Z})|}$ for each step $i$, where $|\mathrm{Dim}(\mathcal{Z})|$ denotes the number of dimensions of the embedding state. We present the results on 10 of the ensemble model in Figure 12. We found that $y_i$ in different dynamics models are almost the same at the first step and become separable after $4 - 5$ steps. This result reveals that different dynamics models are distinguished by $z$. Besides, $y$ are approximately divided into several groups. Hence, the context $z$ is informative and has discovered the environment-specific information.

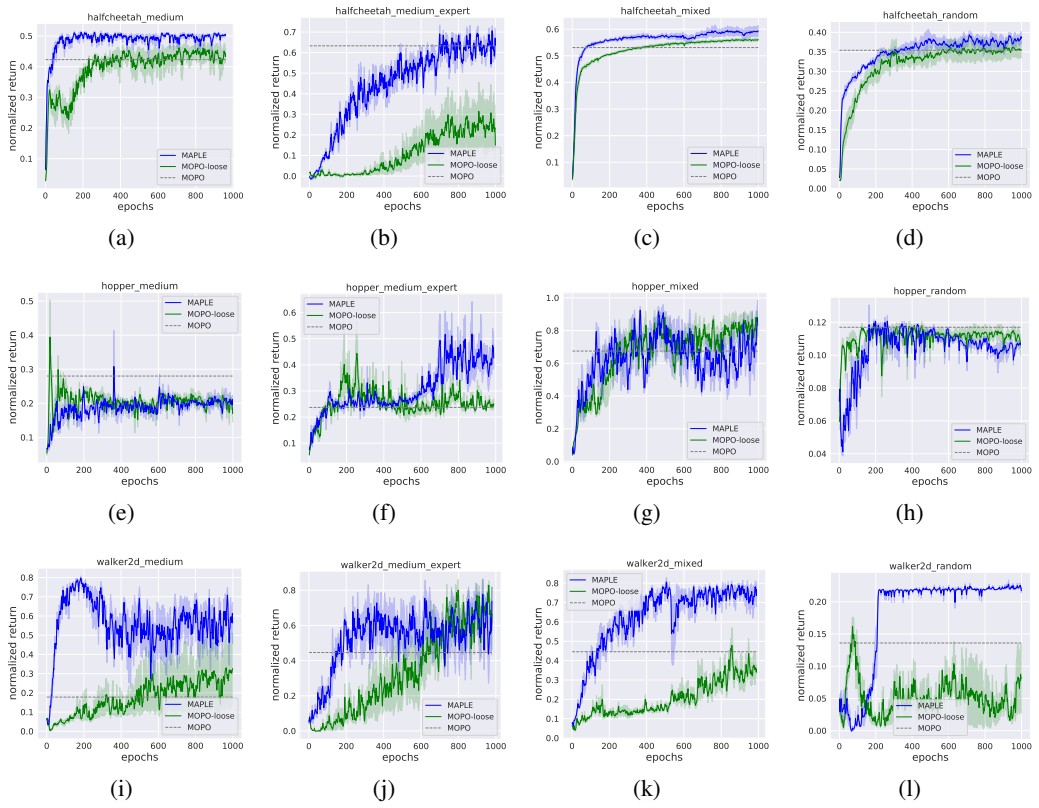

Figure 11: The learning curves of MAPLE, MOPO, and MOPO-loose in mujoco tasks. The solid curves are the mean reward and the shadow is the standard error of three seeds.

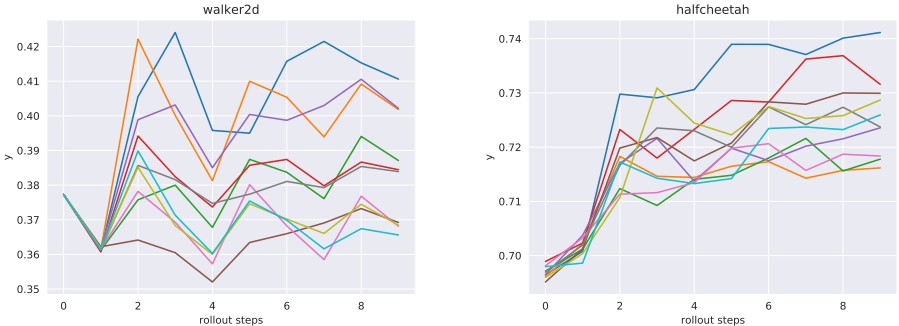

Figure 12: Hidden state visualization in 10 learned dynamics models.

We also sample trajectories for 1000 time-step in the deployment environment. We found that the curves of $y$ oscillate within a region. In HalfCheetah-mixed, the range is around $[6.9, 7.6]$. In Walker2d-mixed, the range is around $[4.0, 4.2]$. The $z$ in the deployment environment are not constant, conversely, they are continuously changing. The changing range is approximately in the range of the converged value in the learned dynamics models. This result implies that the deployment environment could be a combination of the learned dynamics, and thus the context variables in the deployment environment could be all possible values that have appeared in the learned dynamics.

Finally, to further study the behavior of $\Phi$ in the deployment environment, we try to sample trajectories in the deployment environment and disturb the hidden state of the RNN at time step $200$, $400$, and $600$. The context variables are robust to the disturbance. When the disturbance is injected, $y_i$ will

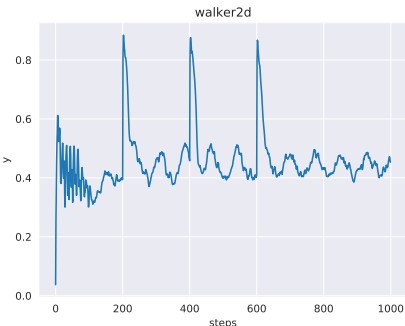
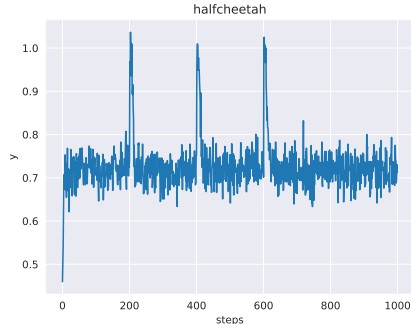

Figure 13: Hidden state visualization in deployment environments. At time step 200, 400, and 600, the hidden state of the RNN is diturbed.

converge back to the region before the disturbance within 10 time steps. Thus, we also believe the predicted contexts are stable and robust.

### F.6    MAPLE with other meta-learning techniques

In MAPLE, meta RL is adopted to solve the model-based offline RL problem. We have conducted the experiments of MAPLE instantiated with the online system identification (OSI) method. In fact, other techniques, e.g., VariBAD [11], can also be integrated into MAPLE. We also implemented VariBAD in MAPLE. We name the new combined method *VariBAD-MAPLE*. In VariBAD-MAPLE, we implement MAPLE with additional auxiliary tasks of the state and reward reconstruction and KL divergence minimization between the inferred $z$ and a prior Gaussian distribution $\mathcal{N}(0, 1)$. All of the other techniques including the truncated horizon and reward penalty in MAPLE are also used in VariBAD-MAPLE. The comparison result is presented in Table 2. We name the MAPLE method instantiated with OSI OSI-MAPLE to distinguish it from VariBAD-MAPLE.

Table 2: Comparisons of different variants of MAPLE.

| Environment | Dataset | OSI-MAPLE | VariBAD-MAPLE | VariBAD | SAC | MOPO |
|---|---|---|---|---|---|---|
| Walker2d | random | $21.7 \pm 0.3$ | $\mathbf{21.8 \pm 0.2}$ | $4.47 \pm 1.9$ | $4.1$ | $13.6 \pm 2.6$ |
| Walker2d | medium | $56.3 \pm 10.6$ | $\mathbf{81.1 \pm 1.2}$ | $4.5 \pm 0.6$ | $0.9$ | $11.8 \pm 19.3$ |
| Walker2d | mixed | $\mathbf{76.7 \pm 3.8}$ | $54.2 \pm 8.7$ | $10.8 \pm 2.4$ | $3.5$ | $39.0 \pm 9.6$ |
| Walker2d | med-expert | $\mathbf{73.8 \pm 8.0}$ | $70.0 \pm 16.2$ | $-0.1 \pm 0.0$ | $-0.1$ | $44.6 \pm 12.9$ |
| HalfCheetah | random | $38.4 \pm 1.3$ | $\mathbf{41.2 \pm 1.1}$ | $37.8 \pm 0.2$ | $30.5$ | $35.4 \pm 1.5$ |
| HalfCheetah | medium | $\mathbf{50.4 \pm 1.9}$ | $50.4 \pm 3.4$ | $22.4 \pm 3.7$ | $-4.3$ | $42.3 \pm 1.6$ |
| HalfCheetah | mixed | $\mathbf{59.0 \pm 0.6}$ | $56.7 \pm 0.5$ | $42.2 \pm 5.7$ | $-2.4$ | $53.1 \pm 2.0$ |
| HalfCheetah | med-expert | $63.5 \pm 6.5$ | $\mathbf{64.9 \pm 6.4}$ | $-0.2 \pm 0.5$ | $1.8$ | $63.3 \pm 38.0$ |

As can be seen in VariBAD, just applying a meta-RL algorithm does not work well in the model-based offline RL domain compared with MAPLE. However, compared with the vanilla SAC algorithm, VariBAD can reach similar or better performance in most of the tasks. In the task of HalfCheetah-random, VariBAD can be even better than MOPO. However, by both considering the issues caused by the approximated dynamics models and the strategy of exploiting new samples obtained during deployment, VariBAD-MAPLE can also reach a significantly better performance than MOPO. Compared with the original implementation of OSI-MAPLE, VariBAD-MAPLE can do better than OSI-MAPLE in Walker2d-medium and HalfCheetah-random but worse than OSI-MAPLE in Walker2d-mixed and HalfCheetah-mixed.

The results again demonstrate the effectiveness of the paradigm of MAPLE. The results also inspire us that paying attention to the connection between the meta-RL techniques in sim2real and model-based offline-RL domains is valuable for model-based offline RL to develop more robust policy learning algorithms.

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

Table 3: Results on MuJoCo tasks. Each number is the normalized score proposed by Fu et al. [10] of the policy at the last iteration of training, $\pm$ standard deviation. We group the experiments by the type of task and the value of $m$ and bold the top-3 scores for each group.

| setting | | | performance | | |
|---|---|---|---|---|---|
| $m$ | $H$ | $\lambda$ | walker-medium | halfcheetah-mixed | hopper-medium-expert |
| | | 1.0 | $31.86 \pm 20.22$ | $59.69 \pm 1.55$ | $22.89 \pm 0.06$ |
| | 5 | 0.25 | $\mathbf{55.81 \pm 8.73}$ | $57.75 \pm 1.59$ | $25.58 \pm 2.12$ |
| | | 0.05 | $31.97 \pm 7.61$ | $58.51 \pm 1.48$ | $24.97 \pm 5.25$ |
| | | 1.0 | $8.69 \pm 0.26$ | $\mathbf{62.19 \pm 0.24}$ | $\mathbf{45.01 \pm 5.18}$ |
| 7 | 10 | 0.25 | $\mathbf{37.70 \pm 16.82}$ | $58.84 \pm 2.05$ | $\mathbf{43.59 \pm 13.20}$ |
| | | 0.05 | $\mathbf{41.50 \pm 13.13}$ | $\mathbf{61.83 \pm 1.58}$ | $27.02 \pm 0.75$ |
| | | 1.0 | $7.80 \pm 0.44$ | $60.80 \pm 0.73$ | $\mathbf{38.11 \pm 1.94}$ |
| | 40 | 0.25 | $6.11 \pm 0.39$ | $61.59 \pm 1.56$ | $26.95 \pm 0.80$ |
| | | 0.05 | $4.23 \pm 1.80$ | $\mathbf{62.00 \pm 1.55}$ | $29.33 \pm 0.19$ |
| | | 1.0 | $5.37 \pm 2.23$ | $58.10 \pm 1.06$ | $24.92 \pm 1.22$ |
| | 5 | 0.25 | $\mathbf{78.14 \pm 1.12}$ | $59.06 \pm 0.53$ | $24.99 \pm 0.80$ |
| | | 0.05 | $\mathbf{76.26 \pm 0.97}$ | $59.45 \pm 0.56$ | $24.82 \pm 1.30$ |
| | | 1.0 | $33.37 \pm 16.61$ | $60.86 \pm 0.65$ | $\mathbf{49.89 \pm 3.83}$ |
| 20 | 10 | 0.25 | $\mathbf{56.30 \pm 10.60}$ | $59.04 \pm 0.60$ | $\mathbf{41.93 \pm 4.93}$ |
| | | 0.05 | $14.91 \pm 7.06$ | $\mathbf{63.18 \pm 1.85}$ | $\mathbf{43.51 \pm 5.86}$ |
| | | 1.0 | $30.50 \pm 15.56$ | $62.05 \pm 0.17$ | $33.98 \pm 0.10$ |
| | 40 | 0.25 | $22.53 \pm 8.35$ | $\mathbf{68.39 \pm 0.27}$ | $36.48 \pm 2.09$ |
| | | 0.05 | $5.36 \pm 1.33$ | $\mathbf{67.72 \pm 0.36}$ | $34.59 \pm 0.31$ |
| | | 1.0 | $12.98 \pm 1.29$ | $58.01 \pm 0.16$ | $28.15 \pm 2.29$ |
| | 5 | 0.25 | $\mathbf{82.61 \pm 1.38}$ | $59.24 \pm 0.92$ | $26.23 \pm 0.16$ |
| | | 0.05 | $\mathbf{54.70 \pm 16.65}$ | $60.79 \pm 1.20$ | $23.45 \pm 0.63$ |
| | | 1.0 | $41.20 \pm 18.24$ | $61.94 \pm 0.27$ | $\mathbf{55.21 \pm 8.42}$ |
| 50 | 10 | 0.25 | $\mathbf{79.35 \pm 0.64}$ | $59.14 \pm 0.89$ | $\mathbf{46.52 \pm 6.89}$ |
| | | 0.05 | $53.79 \pm 8.42$ | $\mathbf{63.88 \pm 1.39}$ | $34.88 \pm 3.59$ |
| | | 1.0 | $8.60 \pm 0.03$ | $61.27 \pm 1.38$ | $\mathbf{38.55 \pm 0.45}$ |
| | 40 | 0.25 | $6.17 \pm 0.37$ | $\mathbf{67.66 \pm 0.66}$ | $30.39 \pm 0.16$ |
| | | 0.05 | $6.52 \pm 0.09$ | $\mathbf{69.00 \pm 0.21}$ | $33.83 \pm 2.91$ |
| MOPO | | | $17.8 \pm 19.3$ | $53.1 \pm 2.0$ | $23.7 \pm 6.0$ |

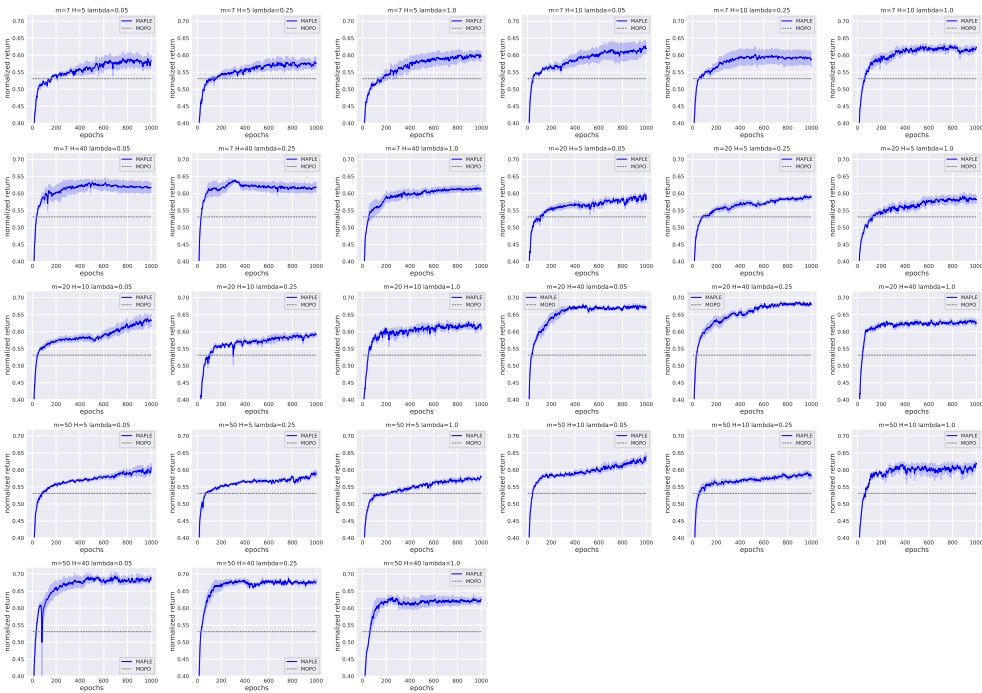

Figure 14: Illustration of hyper-parameter analysis on Halfcheetah-mixed. The solid curves are the mean reward and the shadow is the standard error of three seeds.

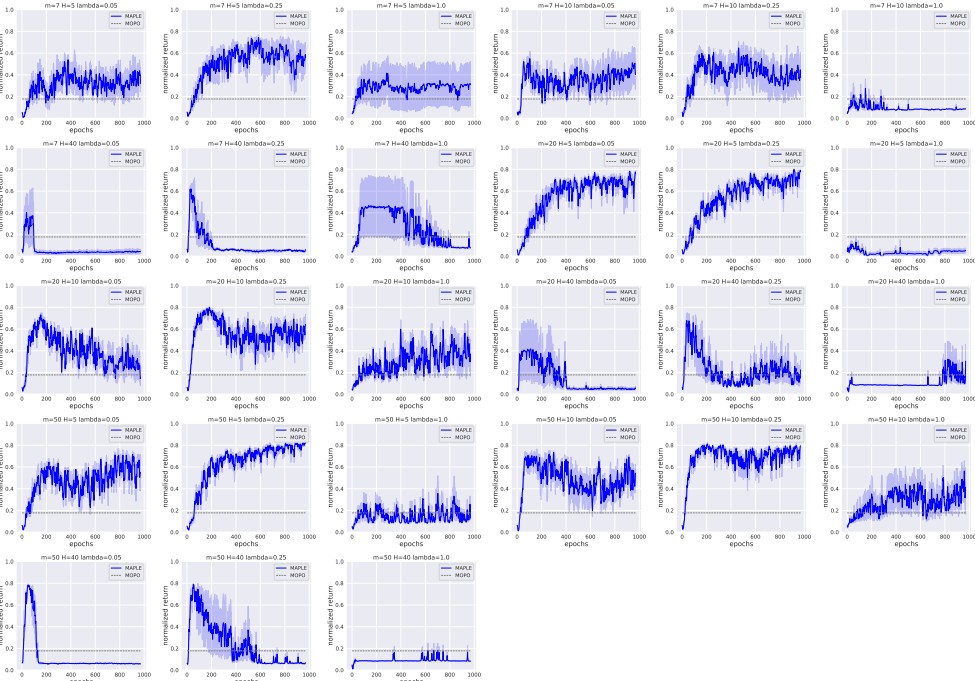

Figure 15: Illustration of hyper-parameter analysis on Walker2d-medium. The solid curves are the mean reward and the shadow is the standard error of three seeds.

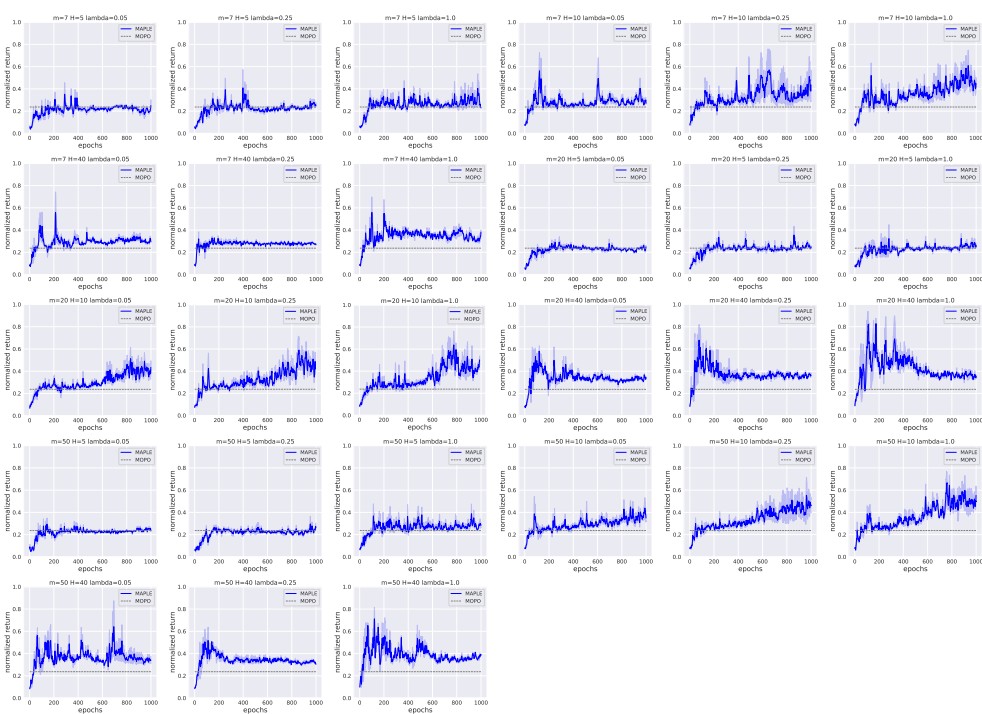

Figure 16: Illustration of hyper-parameter analysis on Hopper-medium-expert. The solid curves are the mean reward and the shadow is the standard error of three seeds.

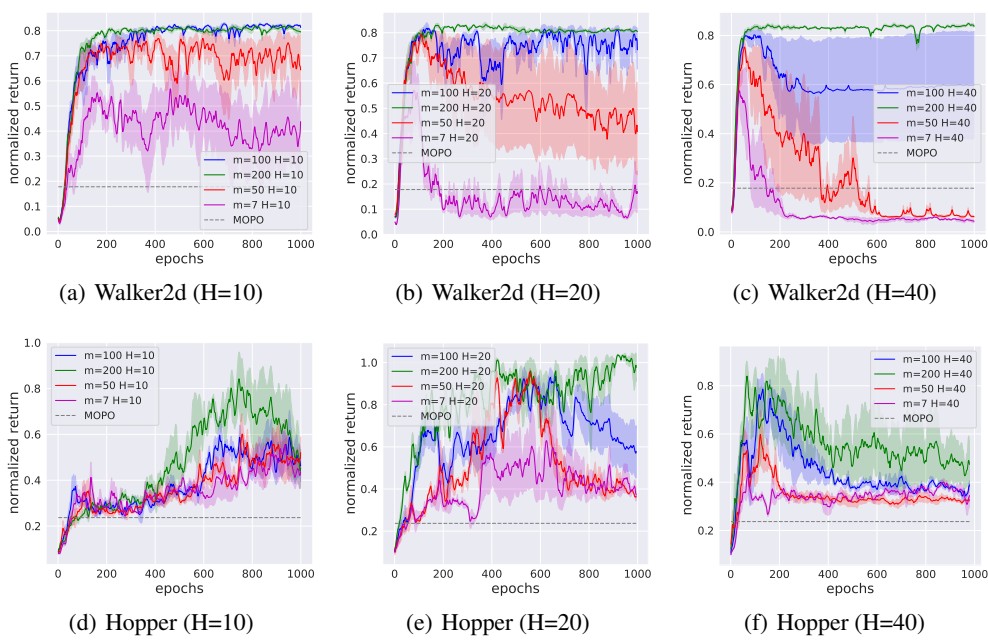

Figure 17: The learning curves of MAPLE with different hyper-parameters $m$ and $H$. The solid curves are the mean of normalized return and the shadow is the standard error.

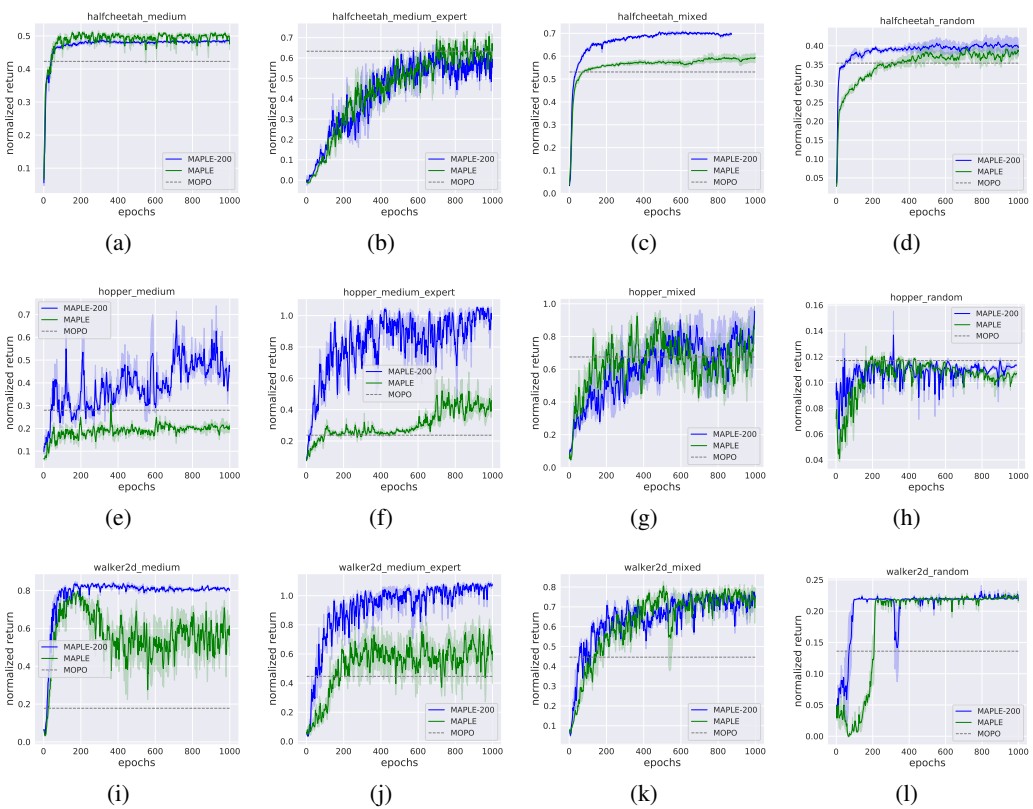

Figure 18: Learning curves of MAPLE-200, MAPLE and MOPO in mujoco tasks. The solid curves are the mean reward and the shadow is the standard error of three seeds.