# OpenReview forum: "Offline Model-based Adaptable Policy Learning"
_NeurIPS.cc/2021/Conference — NeurIPS 2021 Poster_

### Official Review · Reviewer_3man · 2021-06-29

**Rating:** 6
**Confidence:** 5

**Summary:**

This paper proposes to learn adaptable policies in an offline RL setting. One of the limitations of offline RL methods is decision-making in the out-of-support region. This paper tries to leverage existing meta-learning algorithms to learn a latent code z from trajectories as a context variable. Then, a context-conditioned policy would take in both states and context variables for generating actions. They experimentally show this method outperform previous methods on standard offline RL benchmarks.

**Limitations And Societal Impact:**

Mujoco locomotion tasks are standard for offline RL, but in D4RL, there are many more meaningful data that can be used to test the proposed method. It usually remain unclear what happened in the environment while using locomotion tasks as a testbed. For example, except for seeing improved scores, it is unclear to me what has been improved what has been not. Is it really out-of-support? Is context-variable z informative? Can you visualize z?

MAPLE-200 only helps in 2/3 of the tasks. This seems a bit contradictory to the theory part.




**Main Review:**

The paper considers a novel setting wherein the deployment time, the learned policy might need to adapt to have the best performance. They borrow the techniques from meta-learning to learn context-aware policies that can adapt.  The paper is well written for clearly explaining the motivation, intuition, and practical considerations. The paper would contribute toward realistic offline RL methods.

**Time Spent Reviewing:**

2

---

> ### Author Response · Authors · 2021-08-09
> **Official Response to Reviewer 3man**
>
> Thanks for your appreciation of our paper and your constructive comments on it. Our answers to your questions are as follows:
>
> **Q1: Mujoco locomotion tasks are standard for offline RL, but in D4RL, there are many more meaningful data that can be used to test the proposed method. It usually remain unclear what happened in the environment while using locomotion tasks as a testbed. It is unclear what has been improved what has been not. Is it really out-of-support? Is context-variable z informative?  Can you visualize z?**
>
> In MuJoCo tasks, it is indeed hard to see what happened exactly. But we have tried our best to make it clear, which can be seen in the Appendix: In Sec. B, we design a toy environment to make a comparison of step-by-step behaviors among the policies learned by different algorithms. The results show that the adaptable policy learned with the meta-RL technique does the same "probe-reduce" behavior as we excepted in Sec. 4.1.
> Besides, in Sec. F.3, we regard $H^*$ as a metric that how far can the adaptable policy make robust decisions in out-of-support regions,
> and regards the size of ensemble dynamics models $m$ as the coverage in out-of-support regions via the dynamics models set.
> The results in Fig. 10 show that by covering more unknown transitions, MAPLE can make robust decisions in further out-of-support regions.
>
>
> **Q2: MAPLE-200 only helps in 2/3 of the tasks. This seems a bit contradictory to the theory part.**
>
> The bold numbers mark the higher methods in terms of the mean value. Indeed, out of the 12 tasks, there are 4 tasks where MAPLE-200 has a lower mean performance than MAPLE. However, when we consider the standard error, we will find that only one of the 12 tasks of MAPLE-200 is significantly lower than MAPLE. The performance of MAPLE-200 in 6/12 of the tasks is significantly higher than MAPLE. That is, when we increase the model size to 200, the performance in almost all tasks will not significantly drop. Furthermore, in half of the cases, the large model size can significantly improve the performance. We believe such a result is enough to support our theory.

---

> > ### Comment · Reviewer_3man · 2021-08-19
> > **Rebuttal Response**
> >
> > After reading the rebuttal, I think my concerns are resolved. I will recommend to accept.

---

> > > ### Author Response · Authors · 2021-08-28
> > > **Thanks for the response**
> > >
> > > We are glad that our response has resolved your concerns. Thanks for your appreciation of the work.

---

> ### Author Response · Authors · 2021-08-15
> **Results of z visualization**
>
>
>
> We have conducted additional experiments in Walker2d-mixed and HalfCheetah-mixed to visualize the $z$. We first randomly select 10,000 states from the offline dataset and rollout the adaptable policy for 10 steps with each of the dynamics models in the ensemble model set.  We record the absolute value of the context-variable $y_i=\frac{\|z_i\|_1}{\text{Dim}\left(\mathcal{Z}\right)}$ for each step $i$, where $\text{Dim}\left(\mathcal{Z}\right)$ denotes the number of dimension of the embedding state.
> We found that $y_i$ in different dynamics models are almost the same at the first step and become separable after $4$--$5$ steps. We plot the results on 10 of the ensemble model in the link: https://postimg.cc/ct8PbrQj and https://postimg.cc/5jYc0kZ3.
>
> This result reveals that different dynamics models are distinguished by $z$. Besides, $y$ are approximated divided into several groups. Hence,  the context $z$ is informative and has discovered the environment-specific information.
>
> Then we sample trajectories for 1000 time-step in the deployment environment. We found that the curve of $y$ oscillating within a region. In HalfCheetah-mixed, the range is around $[6.9, 7.6]$. In Walker2d-mixed, the range is around $[4.0, 4.2]$. The $z$ in the deployment environment are not constant, conversely,  they are continuously changing. The changing range is approximately in the range of the converged value in the learned dynamics models. This result implies that the deployment environment could be a combination of the learned dynamics, and thus the context variables in the deployment environment could be all possible values that have appeared in the learned dynamics.
>
> Finally, to further study the behavior of $\phi$ in the deployment environment, we try to sample trajectories in the deployment environment and disturb the hidden state of the RNN at time step $200$, $400$, and  $600$. The context variables are robust to the disturbance. When the disturbance is injected, $y_i$ will converge back to the region before the disturbance within $10$ time steps.
> Thus, we also believe the predicted contexts are stable and robust.
>
> The figures of the above results are given in the link: https://postimg.cc/PCFzP2GW and https://postimg.cc/R361fK9j.
>
> We will add the results and the visualization figures to our revised paper.

---

### Official Review · Reviewer_yAV4 · 2021-07-11

**Rating:** 4
**Confidence:** 4

**Summary:**

The authors propose a new offline model-based RL method MAPLE. The method relies on training an ensemble of dynamics with the offline data in addition to an environment-context extractor model that maps from dynamics to a latent embedding, which is part of the input to the policy. The authors claim such an environment-context extractor could both eliminate inconsistent dynamics and thus the remaining policy, after the probing stage, is the optimal one w.r.t. the ground truth dynamics during deployment. The experiments show that MAPLE achieves competitive results compared to state-of-the-art offline RL algorithms.

**Limitations And Societal Impact:**

This work does not seem to have a negative societal impact.

**Main Review:**

Strengths:

1. The development of the paper is well-established. Sec. 4.1 provides a good warm-start in the ideal situation, and the theory result gives good intuition on the tradeoffs and algorithm designs. Though there are still gaps between the transitions that could be fixed (please see below).

2. The algorithm is clearly presented, along with the network architecture provided in the appendix. Seems easy to reproduce the results.

3. The empirical results seem promising. The algorithm achieves competitive results compared to the most state-of-the-art offline reinforcement learning algorithms.

Weakness:

1. The definition of the optimal environment-context extractor $\phi^{\ast}$ (in line 194) seems quite strong. The current definition imposes a strong assumption that there exists a single policy that is optimal for all dynamics models. The assumption could be more  reasonable if it is $\exists \phi^{\ast}, \forall \hat{\rho} \in \mathcal{T}, \exists \pi_{\phi^{\ast}} \in \Pi, J_{\hat{\rho}}(\pi_{\phi^{\ast}}) = max_{\pi}  J_{\hat{\rho}}(\pi)$. This is also because the paper seems to treat the policy and the environment-context extractor as separate classes, and it seems more realistic to assume a global optimal $\phi$ but not policy $\pi$ that is optimal for every dynamics model (as Eq.1 suggests, the best we can do is to seek for a policy with maximum expected returns on the distribution of the dynamics models).

2. The introduction to the set of dynamics $\mathcal{T}$ seems abrupt. At the first glance, it could be hard to follow why one suddenly needs to optimize the policy over a set of dynamics since the results in the previous section assume a convergence to a single policy/dynamics at the end. It could be better if an explanation is provided here for the dynamics model set (e.g., introduce stochasticity while all models are deterministic, or model ensemble, etc.).

3. The reviewer is not fully convinced of the "probing" ability of the environment-context extractor. Across the algorithm there is actually no dynamics-related signal while training $\phi$. As described in the paper, "If the optimal actions in the same state are conflicting, to increase the performance of objective, Equation (1), the policy gradient has to backpropagate from $\pi$ to $\phi$" (line 223-224). Since there is no guarantee that the learned dynamics could extrapolate accurately in out-of-support regime, training $\phi$ purely based on the returns could also make $\phi$ exploit the errors of the models. For example, even if the dynamics of the deployment environment is included in the dynamics set. If all dynamics are consistent up to timestep $i$ but there exists a dynamics that induces a larger expected return than deployment dynamics then the policy, together with $\phi$, will more likely select the optimal action of that particular environment, other than the optimal action of the deployment environment.

Originality:

This work is an incremental work that improves on current model-based offline reinforcement learning algorithms that combine ensemble model learning and pessimism.

Clarity/Quality:

Most of the paper is well written. Some of the technical details could be better explained (as suggested in the weakness section).

Significance:

The major contribution of the work is the empirical competitiveness of the proposed algorithm. The reviewer is not fully convinced by the technical contribution of the work (namely, the technical soundness of the environment-context extractor model).


**Time Spent Reviewing:**

9

---

> ### Author Response · Authors · 2021-08-09
> **Official Response to Reviewer yAV4**
>
> Thanks for your constructive comments on our paper. Our answers to your questions are as follows:
>
> **On the Originality: This work is an incremental work that improves on current model-based offline reinforcement learning algorithms that combine ensemble model learning and pessimism.**
>
> We do not agree that this work is incremental. As far as we know, previous model-based offline RL algorithms study mainly learned conservative policies that stay close to well-support regions. Our work focuses on learning a meta-policy that generalizes to out-of-support regions, which is a new direction to improve the performance of RL policies from offline data.
>
>
> **Q1 The definition of the optimal environment-context extractor $\phi^\star$ (in line 194) seems quite strong. The current definition imposes a strong assumption that there exists a single policy that is optimal for all dynamics models. The assumption could be more reasonable if it is $\exists\phi^\star, \forall \hat{\rho} \in \mathcal{T}, \exists\pi_{\phi^\star}\in\Pi,J_{\hat{\rho}}(\pi_{\phi^\star})=\max_\pi J_{\hat{\rho}}(\pi)$. This is also because the paper seems to treat the policy and the environment-context extractor as separate classes, and it seems more realistic to assume a global optimal $\phi$ but not policy $\pi$ that is optimal for every dynamics model (as Eq.1 suggests, the best we can do is to seek for a policy with maximum expected returns on the distribution of the dynamics models).**
>
> We would like to clarify that there is a misunderstanding on $\Pi$. Note that $\Pi$ is the class of meta-policies, i.e., the policy $\pi_{\phi}$ in the form of $\pi(a_t|z_t, s_t)$. If $\Pi$ was the class of non-meta policies, the reviewer would be correct that the assumption is overly strong because $\pi_{\phi^\star}$ is static and $\max_\pi J_{\hat{\rho}}(\pi)$ changes for every environment. However, since $\Pi$ is the class of meta-policy, $\pi_{\phi^\star}$ is instantiated to different policies in different environments. Thus the assumption is reasonable. We will revise the corresponding part in our paper to make it clear.
>
> **Q2: The introduction to the set of dynamics $\mathcal{T}$ seems abrupt. At the first glance, it could be hard to follow why one suddenly needs to optimize the policy over a set of dynamics since the results in the previous section assume a convergence to a single policy/dynamics at the end. It could be better if an explanation is provided here for the dynamics model set (e.g., introduce stochasticity while all models are deterministic, or model ensemble, etc.).**
>
> By the clarification of Q1, we are not going to converge to a single policy/dynamics, but a meta-policy that can be instantiated to different policies. And the policy is converged to "a single policy" after the extractor $\phi^\star$ distinguish the dynamics and infer a stable $z$. Such a meta-policy is naturally required to be trained over a set of dynamics models, instead of one model. We will revise to make this clear.
>
> **Q3: The reviewer is not fully convinced of the "probing" ability of the environment-context extractor. Across the algorithm, there is actually no dynamics-related signal while training $\phi$. As described in the paper, "If the optimal actions in the same state are conflicting, to increase the performance of objective, Equation (1), the policy gradient has to backpropagate from $\pi$ to $\phi$" (line 223-224). Since there is no guarantee that the learned dynamics could extrapolate accurately in out-of-support regime, training $\phi$ purely based on the returns could also make $\phi$ exploit the errors of the models. For example, even if the dynamics of the deployment environment is included in the dynamics set. If all dynamics are consistent up to timestep $i$ but there exists a dynamics that induces a larger expected return than deployment dynamics then the policy, together with $\phi$, will more likely select the optimal action of that particular environment, other than the optimal action of the deployment environment.**
>
> Still inherited from the misunderstanding in Q1, if a single policy is to be trained, the probing ability would have been absent. However, this work trains a meta-policy that takes not only the state as the input but also a latent variable $z$, and $z$ is the probing result of the environment-context extractor $\phi$. As a result, during the execution of the policy, the policy keeps inferring the latent variable $z$ using $\phi$ and instantiating the meta-policy to be a policy according to $z$.
>
> If all dynamics are consistent up to time step $i$ (which is unlikely due to the random initialization of the model), the environment-context extractor cannot distinct environments up to the time step. Then the policy will select the action $a_i$ with the largest expectation return on the dynamics models. However, as long as the dynamics are inconsistent (otherwise all dynamics equal the deployment environment) after the time step $i$, the environment-context extractor can distinct the dynamics. For example, after getting the $s_{i+1}$ from the deployment environment, the tuple $(s_i, a_i, s_{i+1})$,  which are put into $\phi^\star$, is used to distinct the dynamics and reduce the number of consistent dynamics. At the next time when the agent reaches the same state $s$, where $s=s_i$, the optimal action would  be chosen based on the remaining  dynamics models instead of selecting the $a=a_i$ again.

---

> > ### Comment · Reviewer_yAV4 · 2021-08-25
> > **Response to the authors**
> >
> > The reviewer appreciates the authors for their comments.
> >
> > Q1: The current training procedure (i.e., training $\pi$ and $\phi$ jointly with the same objective) still makes the current assumptions overly strong. In fact, one can think of a new function class $\Pi' = \Pi \times \Phi$, where $\Phi$ is the function class of the extractor $\phi$. For simplicity, one can even think of such function class $\Pi'$ as a family of recurrent networks (just for the latent $z$'s) but can also think of the process as starting with $z_0 = 0$ and rolling out with the output $z_t$ at each time. Thus the assumption is saying that there is a $\textbf{static}$ policy in $\Pi$ that is the best policy in all dynamics. Again this is inferred from the fact that $\pi$ and $\phi$ are trained jointly with the same objective.
> >
> > Q3: the issue of model exploitation is still unclear. Again since the only training objective is to maximize the expected returns, and there is no training signal that is actually relevant to the dynamics (for training $\phi$), it's still unclear how is the proposed algorithm deals with the example that the reviewer provides other than exploiting model errors.

---

> > > ### Author Response · Authors · 2021-08-28
> > > **Response to Reviewer yAV4**
> > >
> > >
> > > Thanks for the detailed response. We still think that there is a misinterpretation of the work.
> > >
> > > An intuitive understanding of the meta-policy is that, as the input is augmented by the environment context, it actually contains many different static policy models by choosing different contexts, rather than one static policy model.
> > > Moreover, the context is extracted by the extractor from a segment of the trajectory, which is to identify the dynamics. Please note that the learning objective of the context extractor is to MAXIMIZE the return of the CHOSEN POLICY in EACH of the environment MODELS SIMULTANEOUSLY. This learning objective naturally drives the extractor to learn how to identify different dynamics. You may also imagine the extractor as a classifier of the environment, and its classification result is to choose the best policy.
> > >
> > > Now we further clarify the process of training based on the reviewer's definition: $\Pi'$, $\Pi$, and $\Phi$. In particular, we have an environment-context extractor $\phi \in \Phi$, which takes a partial trajectory $\tau_{0:t}$  as inputs and output $z_t$ for each timestep. **We have a meta-policy class $\Pi$ in which take $z$ and $s$ as inputs and output action $a$ and a non-meta policy class $\Pi_s$ which takes $s$ as inputs and output action $a$.**
> > > We define the optimal environment-context extractor $\phi^*$ which satisfies:
> > > $$
> > > \exists \pi_a^* \in \Pi, \forall \hat \rho \in \mathcal{T}, J_{\hat \rho} (\pi_a^* \circ \phi^*) = \max_{\pi \in \Pi_s} J_{\hat \rho}(\pi)
> > > $$
> > > where $\pi_a$ denotes an adaptable meta-policy which satisfy $\pi_a^*\in \Pi$, and $\pi_a \circ \phi$ is a composite function: $\pi_a \circ \phi:=\pi_a(a_t|s_t, \phi(z_t|\tau_{o:t}))$ and $\pi_a \circ \phi \in \Pi'$.
> > >
> > > In this formulation, our objective is:
> > > $$
> > > \phi^*, \pi_a^* = \arg \max_{\phi, \pi_a} \mathbb{E}_{\rho \sim \mathcal{T}} [ J_\rho (\pi_a \circ \phi)].
> > > $$
> > >
> > > **The objective of maximizing the expectation return value on the distribution of the dynamics models is enough to find an optimal adaptable policy for all of the dynamics models.**
> > > For different dynamics models $\rho \sim \mathcal{T}$, the transition functions are different. Therefore, even in deterministic environments, with the same action sequence and initial state, the partial trajectories $\tau$ are different at least in some timesteps (otherwise all dynamics are equal to the deployment environment).
> > > To maximize the expectation return value among different dynamics models, the policy has to take different actions $a$ in the same state $s$, since the actions with optimal expectation return value are different among dynamics models (otherwise, it is unnecessary to distinguish the dynamics models). Since there is only a global adaptable policy $\pi_a(a|s,z)$, when the optimal actions in the same state are conflicting, the policy gradient has to backpropagate from $\pi_a$ to $z$, then update the parameters of $\phi$. To output different representations of $z$, $\phi$ has to exploit the difference of the input trajectories among the dynamics models. After that, the adaptable policy can make different actions on the state $s$ based on the differences of the input trajectories to increase the expectation return value on the distribution of the dynamics models.
> > > The reviewer said "there is no training signal that is actually relevant to the dynamics" **but the above process does give the signal related to the dynamics models**: For the trajectories $\{\tau_{0:t}\}$ sampled from different dynamics models, faced to the same $s_{t}$ which has to take different actions among the dynamics models for maximizing the expectation return value, the policy gradient has to give different gradient signal among the trajectories from $\pi$ to $z$ to update $\phi$.  Besides, **the process can not be simplified to "starting with $z=0$ and rolling out with the output $z_t$ at each timestep $t$"**: at the same timestep $t$, $z_t$ is varied since the dynamics models to rollout are different and thus the trajectory $\tau_{0:t}$ are different and the optimal action are different.
> > >
> > >
> > >
> > >
> > > We again clarify that the reviewer's definition of the optimal $\phi^*$, that is
> > > $$
> > > \exists \phi^* \in \Phi, \forall \hat \rho \in \mathcal{T}, \exists \pi_a^* \in \Pi, J_{\hat \rho} (\pi_a^* \circ \phi^*) = \max_{\pi \in \Pi_s} J_{\hat \rho}(\pi),
> > > $$ is **incorrect**. As discussed above, it **is the global adaptable policy $\pi_a$** that guides $\phi$ to learn representations. Should we construct an independent adaptable policy $\pi_a^{\rho}$ for each dynamics model $\rho$, the independent parameters of the policies **would have directly given the ability to take different action given the same state**. In this case, we can not guarantee that the policy gradient has to backpropagate from $\pi_a$ to $z$ to update $\phi$. This speaks for the necessity of a global policy $\pi_a$ and its role in updating in $\phi$.
> > >
> > >
> > > We again clarify the reviewer's example, that is: "even if the dynamics of the deployment environment is included in the dynamics set. If all dynamics are consistent up to timestep $i$ but there exists a dynamics that induces a larger expected return than deployment dynamics then the policy, together with $\phi$, will more likely select the optimal action of that particular environment, other than the optimal action of the deployment environment.”
> > > As mentioned in the previous response, in this case, if all dynamics are consistent up to time step $i$, then the policy indeed selects the action $a_i$ with the largest expectation return value on the dynamics models.
> > > It is inevitable since the environment-context extractor cannot distinguish environments through the trajectories $\{\tau_{o:i}\}$ up to the time step $i$.
> > > However, it will not exploit the model errors. As long as the dynamics are inconsistent (otherwise all dynamics are equal to the deployment environment) after the time step $i$, the environment-context extractor can distinguish the dynamics and adjust the behavior of policy through the received different trajectories $\{\tau\}$. For example, after getting the $s_{i+1}$ from the deployment environment, the tuple $(s_i, a_i, s_{i+1})$,  which is put into $\phi^\star$, is different from some of the dynamics models for training. The tuple then is used to distinguish the dynamics and output different $z$ via the extractor $\phi^*$.
> > > **At the next time $k~(k>i+1)$, the agent reaches the same state $s_k$, where $s_k=s_i$, the optimal action will be chosen based on the new input of $z_k$ instead of selecting the $a_k=a_i$ again. In fact, the action taking in timestep $i$ belongs to the process that we call "probing", which given an action that could be incorrect to collect real samples to help us identify the correct representation of $z$.** The probing process might lead to performance degeneration, which has been discussed in Theorem 1 but will not make the agent repeatedly select the incorrect actions caused by the error of dynamics models.
> > >
> > >
> > > In Sec. B of the Appendix, we design a toy environment to make a comparison of step-by-step behaviors among the policies learned by different algorithms. The results show that the adaptable policy learned with the meta-RL technique does the same "probe-reduce'' behavior as we excepted in Sec. 4.1. We hope that the response and the experiment can resolve the misinterpretation. Looking forward to the reviewer's response if there is still any confusion about this work.

---

### Official Review · Reviewer_yJkf · 2021-07-16

**Rating:** 6
**Confidence:** 4

**Summary:**

Conservatism, i.e. avoiding out-of-distribution actions (and states), is a major theme in offline RL because of the need to avoid erroneous extrapolation beyond the data. This paper takes an alternative approach and instead trains a policy that adapts when deployed beyond the data distribution. To do this, a context encoder RNN is trained to produce latent codes given the episode history, and the encoder and policy are jointly optimized to maximize average performance across a large ensemble of pretained dynamics models. The algorithm, offline Model-based Adaptable Policy LEarning (MAPLE), is shown to achieve strong results on the MuJoCo tasks from the D4RL benchmark.

**Limitations And Societal Impact:**

They discuss the limitations thoroughly (which is appreciated), but not the societal impacts.

**Main Review:**

The algorithm is largely a combination of the ideas of MOPO and PEARL, but the combination is novel to my knowledge and well-motivated in the setting discussed. It is different than PEARL in that the meta-learning is happening in a model-based fashion across different dynamics models. However in that sense it is conceptually related to MB-MPO (https://arxiv.org/pdf/1809.05214.pdf), an algorithm which meta-learns a policy via MAML to quickly adapt to any model in the ensemble. I think MB-MPO should be cited and discussed in related work.

The theorem appears correct, and perhaps useful for high-level intuition regarding when/why the algorithm succeeds, but the result is not very informative in my opinion because all the heavy lifting is being done by the unknown coefficient $N_m$, for which no further analysis is presented.

The results on the D4RL MuJoCo tasks look good! One suggestion you may consider to further improve the paper is to experiment with tasks that require leaving the behavior distribution in order to succeed. This was one of the motivations of MOPO, and the authors developed two tasks specifically for this purpose (cheetah-jump and ant-angle).

The writing of the paper is clear and I have no significant issues with it.

**Time Spent Reviewing:**

2

---

> ### Author Response · Authors · 2021-08-09
> **Official Response to Reviewer yJkf**
>
> Thanks for your constructive comments on our paper. We will amend our paper for better clarity. Our answers to your questions are as follows:
>
> **Q1:  it is conceptually related to MB-MPO (https://arxiv.org/pdf/1809.05214.pdf), an algorithm which meta-learns a policy via MAML to quickly adapt to any model in the ensemble. I think MB-MPO should be cited and discussed in related work.**
>
> Thank you for pointing out this reference. We will add this discussion in our paper. Meanwhile, we notice that the MAML technique requires to re-train the model online, taking extra online data, while the online system identification avoids the re-train and thus is purely offline.
>
> **Q2: The theorem appears correct, and perhaps useful for high-level intuition regarding when/why the algorithm succeeds, but the result is not very informative in my opinion because all the heavy lifting is being done by the unknown coefficient Nm, for which no further analysis is presented.**
>
> Intuitively, $N_m$ should depend on the size of state/action space and the type of tasks.  However, we have tried but do have not a good idea to model $N_m$ further. This analysis can be reserved for future work.
>
> **Q3: The results on the D4RL MuJoCo tasks look good!  One suggestion you may consider to further improve the paper is to experiment with tasks that require leaving the behavior distribution in order to succeed. This was one of the motivations of MOPO, and the authors developed two tasks specifically for this purpose (cheetah-jump and ant-angle).**
>
>
> Thank you for your kind advice. We will include the suggested experiments. Since MAPLE also adopts the model ensemble with uncertainty, we believe it will achieve at least as well as MOPO. Moreover, we found that the details of the original experiments are not all available. The hyperparameters and the offline dataset for the two environments are missing in the open-sourced codes. Thus the experiment will take some time.

---

> > ### Comment · Reviewer_yJkf · 2021-09-01
> > **Response to authors**
> >
> > Thank you for addressing my comments! My score will remain as-is.

---

> ### Author Response · Authors · 2021-08-15
> **Experiment Results of HalfCheetah-jump and Ant-angle**
>
> We have tried to reimplement HalfCheetah-jump and Ant-angle and list the results of HalfCheetah-jump as below:
>
>
>
> |      |    MOPO  |  MOPO (reimplementation)     | MOPO-loose |  MAPLE |
> | ---- | ---- | ---- | ---- | ---- |
> |HalfCheetah-Jump | $4140.6 \pm 88.6$ | $5127.6 \pm 310.5$ | $6085.3 \pm 200.9$ | $\mathbf{6893.9 \pm 143.5}$  |
>
> We train a policy on HalfCheetah via standard SAC and store the entire replay buffers as the offline dataset. The dataset size is 1 million. After that, we reassign the reward with the "jump reward" (the same as the original paper) and train dynamics models and the MAPLE/MOPO policies. We reimplemented the MOPO with the same hyper-parameters as the original paper (model size is 7, rollout horizon is 5, and reward penalty coefficient is 1.0). The results show that our MOPO can reach the same performance as the reported results. We run MAPLE with model size being 20, rollout horizon being 10, and reward penalty coefficient being 1.0. For a fair comparison, we also implement MOPO-loose, which is the MOPO algorithm with the same hyper-parameters as MAPLE. The results of MOPO and MOPO-loose are worse than MAPLE. The result shows that MAPLE possesses a stronger generalization ability than MOPO.
>
>
> We also tried to implement Ant-angle but failed: the angle reward needs the information of the y-axis velocity, which is missed in the state space of the Ant task in the Gym. To reassign the reward in the offline dataset, we have to modify the original state space. However, in the modified Ant task, the performance of the standard SAC algorithm can not reach the same performance as the reported results. The MOPO trained with the offline dataset can also not reach the reported performance.

---

> > ### Comment · Reviewer_yJkf · 2021-09-01
> > **Response to authors**
> >
> > Thank you for running this additional experiment! The performance of MAPLE is indeed good, as one would hope. This provides additional strength to the claim that MAPLE can succeed outside the support of the dataset.

---

> > > ### Author Response · Authors · 2021-09-02
> > > **Thanks for the response**
> > >
> > > Thanks for your response and appreciation of the work!

---

### Official Review · Reviewer_g3xy · 2021-07-19

**Rating:** 6
**Confidence:** 4

**Summary:**

This paper presents Model-based Adaptable Policy LEearning (MAPLE), a model-based offline RL algorithm that learns a policy that can quickly adapt to an unknown environment during execution. Existing offline RL algorithms constrain the policy to visit only in-support regions, but MAPLE aims to model all possible dynamics models for out-of-support regions and try to learn a dynamic-conditioned (context-conditioned) policy. Policies for various uncertain dynamics models are prepared first, and gradually filter out policies that do not correspond well to transition observations when deployed. However, solving it exactly is intractable, and a practical approximation is presented, which uses an ensemble dynamics model set and an RNN-based environment-context extractor. Experimental results show that MAPLE outperforms baselines in D4RL benchmarks, showing better generalization ability.


**Ethical Concerns:**

-

**Limitations And Societal Impact:**

Limitations have been discussed in the paper.

**Main Review:**

Overall, the paper is well written and easy to follow. In contrast to existing offline RL algorithms whose computed policy is fixed when deployed, MAPLE can exploit new experiences obtained during test time and can exhibit fast adaptation to the true environment. I found this research direction interesting and novel in the context of offline RL.
- Still, I have some concerns when compared with meta RL, sim2real, domain randomization techniques. It would be great to provide some explanations of what makes MAPLE distinct from those approaches (e.g. applying meta RL whose task distribution was simply replaced with the uniform distribution over the ensemble of the learned dynamics model).
- In the experiments, MAPLE is the only algorithm that uses RNN-based policy and exploits new samples obtained during deployment. Thus, it might be unsurprising that MAPLE outperforms the baseline algorithms. Then, for example, can VariBAD [1] (where the task distribution is defined by the ensemble of learned dynamics models) be comparable with MAPLE? If so, it would be great to see the comparison results.
- Why the reward penalty $-\lambda U(s,a)$ is required for MAPLE? What will be the result without such uncertainty penalty?
- In Figure 2, if the transition after taking $\beta$ is totally unknown, we need to consider infinitely many 'stochastic' transition models, rather than considering deterministic transitions only.
- line 101: $p(\tau | \pi)$ -> $p(\tau | \pi, \rho)$

[1] Zintgraf et al., VariBAD: A Very Good Method for Bayes-Adaptive Deep RL via Meta-Learning, 2019


**Time Spent Reviewing:**

5

---

> ### Author Response · Authors · 2021-08-09
> **Official  Response to Reviewer g3xy**
>
> Thanks for your appreciation of our paper and your constructive comments to it. We will revise the mentioned typo and our answers to your questions are as follows:
>
> **Q1: I have some concerns when compared with meta RL, sim2real, domain randomization techniques. It would be great to provide some explanations of what makes MAPLE distinct from those approaches (e.g. applying meta RL whose task distribution was simply replaced with the uniform distribution over the ensemble of the learned dynamics model).**
>
> We would like to emphasize that our contribution is to provide a new paradigm MAPLE for tackling offline model-based RL problems: previous model-based offline-RL studies learned conservative policies that stay close to in-support regions, while this work aims at learning a meta-policy that generalizes to out-of-support regions.
>
> The implementation of MAPLE shares the cross-domain idea with meta-RL, particularly domain randomization and sim2real. In the current implementation, the online system identification (OSI) method in meta RL is employed. Other techniques, e.g., the mentioned VariBAD, can also be integrated into MAPLE. We also implemented VariBAD in MAPLE, of which the results are listed in Q3.
>
> At the same time, meta-RL or sim2real techniques alone cannot well solve the model-based offline-RL problem directly, and there are other issues to be considered: First, the unreliability of approximated dynamics models in unseen regions for training the policy (thus a reward penalty, short-horizon rollout are used in this work); Second, the absence of environment-parameter to generate an environment-task distribution directly (thus ensemble techniques to generate dynamics models should be considered in this work).
>
> By this work, we would like to inspire more researchers in the community to pay attention to the connection between the sim2real RL and model-based offline-RL domains.
>
> **Q2: Why the reward penalty $-\lambda U(s,a)$ is required for MAPLE?  What will be the result without such uncertainty penalty?**
>
>
> While the meta-RL technique can learn a generalizable policy from a set of models, the generalization error is still related to the distance between the target environment and the models. As discussed in Sec 4.3, in the regions too far away from the dataset, the error of dynamics model prediction would be very large thus we might need a much larger size of ensemble models to cover the real one in these regions. Therefore, we still need to constrain each model to be close to the offline data, which is implemented by the reward penalty from the model uncertainty.
>
> In Sec. F.2 of the Appendix, we provide the results of different $\lambda$ values. The results imply that $\lambda$ is necessary, its removal (or a very small value) will injure the performance.
>
>
> **Q3: In the experiments, MAPLE is the only algorithm that uses RNN-based policy and exploits new samples obtained during deployment. Thus, it might be unsurprising that MAPLE outperforms the baseline algorithms. Then, for example, can VariBAD (where the task distribution is defined by the ensemble of learned dynamics models) be comparable with MAPLE?  If so, it would be great to see the comparison results.**
>
> As mentioned above, without considering special issues in model-based offline RL problems, it does not work that directly adopts the meta-RL algorithms into the offline setting by simply replacing the task distribution with the uniform distribution over the ensemble of the learned dynamics model. We have adopted VariBAD directly via the above way as a meta-RL baseline which can exploit new samples obtained during deployment.
>
>  Besides, VariBAD can be integrated into the paradigm of MAPLE as discussed above. We will report the results of this variant, named VariBAD-MAPLE. In particular, we implement MAPLE with additional auxiliary tasks of the state and reward reconstruction and KL divergence minimization between the inferred $z$ and a prior Gaussian distribution $\mathcal{N}(\textbf{0},\textbf{1})$.
>  It should be noticed that VariBAD-MAPLE combines **all of the other techniques** including the truncated horizon and reward penalty mentioned in MAPLE.
>
> The results on VariBAD and VariBAD-MAPLE will be posted as soon as possible.
>
> **Q4 In Figure 2, if the transition after taking $\beta$ is totally unknown, we need to consider infinitely many 'stochastic' transition models, rather than considering deterministic transitions only.**
>
> In the example of Figure 2, we mean the transition after taking $\beta$ is deterministic but unknown. We will revise to clarify this. Therefore, only two models are sufficient to train optimal policies for each situation.

---

> > ### Author Response · Authors · 2021-08-10
> > **Update results on VariBAD and VariBAD-MAPLE**
> >
> > The experiments on VariBAD and VariBAD-MAPLE are posted as below:
> >
> > |   dataset  |  OSI-MAPLE  |   VariBAD-MAPLE   | VariBAD  | SAC | MOPO|
> > | ----            | ---- | ------- |   ---- | ---- |   ---- |
> > |   Walker2d-random   |  21.7 $\pm$ 0.3    |   **21.8 $\pm$ 0.2**  |  4.47  $\pm$ 1.9    |   4.1   |  13.6 $\pm$ 2.6 |
> > |   Walker2d-medium   |  56.3 $\pm$ 10.6    |   **81.1 $\pm$ 1.2**   |  4.5 $\pm$ 0.6   |   0.9   | 11.8 $\pm$ 19.3 |
> > |   Walker2d-mixed   |  **76.7 $\pm$ 3.8**    |    54.2 $\pm$ 8.7  |  10.8 $\pm$ 2.4   |   3.5  |39.0 $\pm$ 9.6  |
> > |   Walker2d-med-expert   |  **73.8 $\pm$ 8.0**    |   70.0 $\pm$ 16.2   |  -0.1 $\pm$ 0.0    |   -0.1   | 44.6 $\pm$ 12.9 |
> > |   HalfCheetah-random   |  38.4 $\pm$ 1.3    |   **41.2 $\pm$ 1.1**   |   37.8 $\pm$ 0.2   |   30.5  | 35.4 $\pm$ 1.5 |
> > |   HalfCheetah-medium   |  **50.4 $\pm$ 1.9**    |   **50.4 $\pm$ 3.4**   |  22.4 $\pm$  3.7   |   -4.3   | 42.3 $\pm$ 1.6 |
> > |   HalfCheetah-mixed   | **59.0 $\pm$ 0.6**  |   56.7 $\pm$ 0.5   |  42.2 $\pm$ 5.7  |   -2.4   | 53.1 $\pm$ 2.0 |
> > |   HalfCheetah-med-expert   |  63.5 $\pm$ 6.5    |   **64.9 $\pm$ 6.4**   |  -0.2 $\pm$ 0.5    |   1.8   | 63.3 $\pm$ 38.0 |
> >
> >
> >
> > The results of OSI-MAPLE, SAC, and MOPO are copied from Tab. 1 in the main body of our paper. OSI-MAPLE is the original implementation of MAPLE in our paper.
> >
> > As can be seen in VariBAD, just applying a meta-RL algorithm does not work well in the model-based offline RL domain compared with MAPLE. However, compared with the vanilla SAC algorithm, VariBAD can reach similar or better performance in most of the tasks. In the task of HalfCheetah-random, VariBAD can be even better than MOPO.
> >
> >
> > As discussed above, the implementation of MAPLE shares the cross-domain idea with meta-RL. We can construct another variant of MAPLE, i.e.,  VariBAD-MAPLE, as another implementation. By both considering the issues caused by the approximated dynamics models and the strategy of exploits new samples obtained during deployment, VariBAD-MAPLE can also reach a significantly better performance than MOPO. Compared with the original implementation of OSI-MAPLE, VariBAD-MAPLE can do better than OSI-MAPLE in Walker2d-medium and HalfCheetah-random but worse than OSI-MAPLE in Walker2d-mixed and HalfCheetah-mixed.
> >
> > The results again demonstrate the effectiveness of the paradigm of MAPLE. The results also inspire us that paying attention to the connection between the meta-RL techniques in sim2real and model-based offline-RL domains is valuable for model-based offline RL to develop more robust policy learning algorithms. We will include the above results in the revised paper.

---

### Author Response · Authors · 2021-09-01
**General Response**

Thanks for the appreciation and constructive comments of the reviewers on our work.

In the discussion period, we added results on a comparison with a pure meta-learning algorithm, an evaluation of task that requires going beyond the behavior distribution, and a visualization of the inferred environment contexts. In particular,
1. We conducted experiments on VariBAD and VariBAD-MAPLE to verify the performance of the pure meta-learning algorithm and demonstrate the effectiveness of the MAPLE framework with different meta-learning implementations (Detailed results can be found in the response to g3xy);
2. We conducted experiments on HalfCheetah-jump to show the performance improvement on the task that requires leaving the behavior distribution (Detailed results can be found in the response to yJkf);
3. We conducted experiments to visualize the inferred environment-context z in the task of Walker2d-mixed and HalfCheetah-mixed. (Detailed results can be found in the response to 3man);

The reviewers’ suggestions on the additional experiments are highly valuable, and we will add these results to our revised paper. We also responded to the reviewers’ other concerns about this work.

In this work, instead of learning conservative policies as all previous works do, we focus on learning a meta-policy that can adapt and make robust decisions in out-of-support regions. The algorithm is easy to implement, and the performance improvement is significant.  Based on the discussion, we believe the proposed MAPLE gives a new direction to solve the offline learning problem in RL and is valuable to inspire more powerful methods and applications in the RL community.

We believe that our responses have addressed the reviewers' concerns about the work. Since there are no additional questions about the work, we sincerely hope the reviewers will consider revising the score of this work (and thanks to reviewer 3man who has already agreed to do so). We also look forward to reviewers’ responses if you have any further concerns about this work.

---

### Decision · Program_Chairs · 2021-09-27

**Decision:**

Accept (Poster)

**Comment:**

Most reviewers agreed that the paper has an interesting new way of performing offline rl (i.e., how to do well in out-of-support regions), and the empirical results on standard benchmark are very promising.